# CausalAdv: Adversarial Robustness through the Lens of Causality

**Yonggang Zhang**[1,2]  **Mingming Gong**[3]  **Tongliang Liu**[4]  **Gang Niu**[5]  **Xinmei Tian**[1]
**Bo Han**[2,†]  **Bernhard Schölkopf**[6]  **Kun Zhang**[7,8]

[1]University of Science and Technology of China  [2]Hong Kong Baptist University

[3]The University of Melbourne  [4]The University of Sydney

[5]RIKEN Center for Advanced Intelligence Project  [6]Max Planck Institute for Intelligent Systems

[7]Carnegie Mellon University  [8]Mohamed bin Zayed University of Artificial Intelligence

## Abstract

The adversarial vulnerability of deep neural networks has attracted significant attention in machine learning. As causal reasoning has an instinct for modeling distribution change, it is essential to incorporate causality into analyzing this specific type of distribution change induced by adversarial attacks. However, causal formulations of the intuition of adversarial attacks and the development of robust DNNs are still lacking in the literature. To bridge this gap, we construct a causal graph to model the generation process of adversarial examples and define the adversarial distribution to formalize the intuition of adversarial attacks. From the causal perspective, we study the distinction between the natural and adversarial distribution and conclude that the origin of adversarial vulnerability is the focus of models on spurious correlations. Inspired by the causal understanding, we propose the *Causal*-inspired *Adv*ersarial distribution alignment method, CausalAdv, to eliminate the difference between natural and adversarial distributions by considering spurious correlations. Extensive experiments demonstrate the efficacy of the proposed method. Our work is the first attempt towards using causality to understand and mitigate the adversarial vulnerability.

## 1 Introduction

The seminal work (Szegedy et al., 2014; Biggio et al., 2013) shows that DNNs are vulnerable to adversarial examples, which consist of malicious perturbations imperceptible to humans yet fooling state-of-the-art models (Krizhevsky et al., 2012; Szegedy et al., 2015; Simonyan & Zisserman, 2015; He et al., 2016). The lack of robustness hinders the applications of DNNs to some safety-critical areas such as automatic driving (Tuncali et al., 2018) and healthcare (Finlayson et al., 2019). Therefore, mitigating the adversarial vulnerability is critical to the further development of DNNs.

Human cognitive systems are immune to the distribution change induced by adversarial attacks because humans are more sensitive to causal relations than statistical associations (Gopnik et al., 2004). Using causal language, causal reasoning can identify causal relation and ignore nuisance factors, i.e., not the cause of labels, by intervention (Pearl, 2009; Peters et al., 2017). As adversarial perturbations are usually imperceptible and make no impact on human decisions (Szegedy et al., 2014; Goodfellow et al., 2015), it is reasonable to assume that the difference between natural and adversarial examples comes from nuisance factors. This is because if task-relevant factors of some samples are changed, these samples will make both humans and DNNs change their decisions. From a causal viewpoint (Zhang et al., 2020a), adversarial attacks can be regarded as a specific type of distribution change resulting from the intervention on the natural data distribution. In summary, causal reasoning has the instinct for analyzing the effect of the intervention caused by adversarial attacks, so it is essential to leverage causality to understand and mitigate the adversarial vulnerability.

---

[†]Corresponding author (bhanml@comp.hkbu.edu.hk).

However, there are two significant problems to overcome before using causality to understand and mitigate the adversarial vulnerability. Firstly, constructing a causal graph is arguably the fundamental premise for causal reasoning (Pearl, 2009; Peters et al., 2017), but how to construct causal graphs in the context of adversarial attacks is still lacking in the literature. Secondly, using causal language to formalize the intuition of adversarial attacks is the key to connect causality and adversarial vulnerability, but it also remains to be solved. These two problems are fundamental obstacles, which prevent us from employing causality to contribute to adversarial learning.

To address these challenges, we first construct a causal graph to model the perceived data generation process where nuisance factors are considered. The constructed causal graph can entail a specific intervention distribution, i.e., the adversarial distribution. Moreover, the causal graph immediately shows that, given inputs, labels are statistically correlated with nuisance factors, which have no cause-effect to labels. The spurious correlation implies that if DNNs fit the conditional association between labels and nuisance factors, their performance on different conditional associations between labels and nuisance factors will change accordingly. Through investigating the distinction between these two distributions induced by nuisance factors, we conclude that *adversarial distributions can be obtained by exploiting conditional associations between labels and nuisance factors, where the conditional association of adversarial distributions is drastically different from that of natural distributions.* Namely, the origin of adversarial vulnerability is the focus of DNNs on the spurious correlation between labels and nuisance factors.

According to the causal perspective, an adversarial distribution is crafted by exploiting a specific conditional association between labels and nuisance factors, this association is drastically different from that of natural distributions. Intuitively, eliminating the difference in such conditional associations between natural and adversarial distributions can promote the performance of models on adversarial distributions. Thus, we propose the *Causal*-inspired *Adv*ersarial distribution alignment method, CausalAdv, to eliminate the difference between these two distributions. Surprisingly, we find that the proposed method shares the same spirits to existing adversarial training (Goodfellow et al., 2015) variants, i.e., Madry (Madry et al., 2018) and TRADES (Zhang et al., 2019). We validate the efficacy of the proposed method on MNIST, CIFAR10, and CIFAR100 (Krizhevsky et al., 2009) datasets under various adversarial attacks such as FGSM (Goodfellow et al., 2015), PGD (Madry et al., 2018), CW attack (Carlini & Wagner, 2017), and AutoAttack (Croce & Hein, 2020). Extensive experiments demonstrate that CausalAdv can improve the adversarial robustness significantly.

Our main contributions are:

- We provide a causal perspective to understand and mitigate the adversarial vulnerability, which is the first attempt towards using causality to contribute adversarial learning.

- To leverage causality to contribute adversarial learning, we solve two fundamental problems. Specifically, we construct a causal graph to model the adversarial data generation process and define the adversarial distribution to formalize adversarial attacks.

- A defense method called causal-inspired adversarial distribution alignment, CausalAdv, is proposed to reduce adversarial vulnerability by eliminating the difference between adversarial distribution and natural distribution. Extensive experiments demonstrate that the proposed method can significantly improve the adversarial robustness.

## 2  A CAUSAL VIEW ON ADVERSARIAL DATA GENERATION

The ability of humans to perform causal reasoning is arguably an essential factor that makes human learning different from deep learning (Schölkopf et al., 2021; Zhang et al., 2020a; Gopnik et al., 2004). The superiority of causal reasoning endows humans with the ability to identify causal relations and ignore nuisance factors that are not relevant to the task. In contrast, DNNs are usually trained to fit the perceived information overlooking the ability to distinguish causal relations and statistical associations. This *shortcut* solution could lead to overfitting to these nuisance factors, which would further result in the sensitivity of DNNs to such factors. Therefore, we propose incorporating causal reasoning to mitigate the sensitivity of DNNs to these nuisance factors.

Before using causal reasoning to analyze adversarial vulnerability, we need to construct a causal graph, as causal graphs are the key for formalizing causal reasoning (Peters et al., 2017). In the

context of adversarial learning, we desire the causal graph by which both the natural and the adversarial distributions can be generated. In addition, the graph is required to reflect the impact of nuisance factors on these two distributions, so that we can investigate the difference in nuisance factors between these two distributions. Consequently, we can formally establish the connection between nuisance factors and adversarial vulnerability. Therefore, we propose constructing a causal graph to model the adversarial generation process where the nuisance factors are considered. One approach is to use causal structure learning to infer causal graphs (Pearl, 2009; Peters et al., 2017), but it is challenging to apply this kind of approach to high-dimensional data. Using external knowledge to construct causal graphs is another approach (Zhang et al., 2013; Tang et al., 2020; Schölkopf et al., 2021). As automatically learning a precise causal graph is out of scope for this work, external human knowledge about the data generation process is employed to construct the causal graph.

Specifically, we construct a causal graph $\mathcal{G}$ to formalize the image data generation process using the following knowledge for analyzing adversarial vulnerability. As there might be a number of different causes of natural data $X$, we propose to divide all the causes into two categories for simplicity. We group content-related causes into one category, called content variable $C$. The rest causes, i.e., nuisance factors, are grouped into another category, called style variable $S$, which is content-independent, i.e., $S \perp\!\!\!\perp C$. This implies that $C \rightarrow X \leftarrow S$. It is noteworthy that, in this paper, we assume that only the content variable is relevant for the task we care about, i.e., $C \rightarrow Y$. Perceived data $\tilde{X}$ are usually composed of perturbations $E$ and natural data $X$. When the perturbation $E$ is designed carefully to fool DNNs, $E$ should be a compound result of $X$ (the object to be perturbed), $Y$ (the reference for the perturbation),

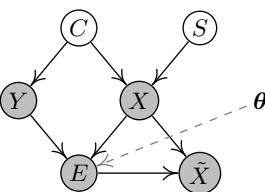

Figure 1: Causal graph of the perturbed data generation process. Each node represents a random variable, and gray ones indicate observable variables, where $C, S, X, Y, E, \tilde{X}, \boldsymbol{\theta}$ are content variable, style variable, natural data, label, perturbation, perturbed data and parameters of a neural network, respectively.

and $\boldsymbol{\theta}$ (the targets affected by the perturbation), e.g., white-box attacks (Goodfellow et al., 2015; Carlini & Wagner, 2017; Moosavi-Dezfooli et al., 2016), which means that $(X, Y, \boldsymbol{\theta}) \rightarrow E$ [1]. Leverage all this background knowledge, we obtain the causal graph $\mathcal{G}$ formalizing the perturbed data generation process, depicted in Fig. 1.

Based on the causal graph, we can define valid interventions (Pearl, 2009; Schölkopf et al., 2021) and the corresponding intervention distributions. Defining valid interventions is equivalent to determining which variables or mechanisms in the causal graph can be intervened. In this work, we consider both hard and soft interventions (Eberhardt & Scheines, 2007; Correa & Bareinboim, 2020) on the perturbation variable $E$. Specifically, we can use a structural causal model to represent the generating mechanism of $E$:

$$E := \mathcal{M}(X, Y, \boldsymbol{\theta}, U_E), \tag{1}$$

where the exogenous variable $U_E$ stands for other indeterminacies, e.g., random start noise used in PGD attack (Madry et al., 2018).

By intervening $E$ in different ways, we can obtain different intervention distributions, which could be either natural or adversarial, over the observed variables. For example, if we do a hard intervention on $E$, i.e., $do(E = \mathbf{0})$ in the graph, the generated data distribution corresponds to the natural distribution $P(X, Y)$. Different perturbations can be obtained by performing soft interventions on $E$, i.e., modifying $\mathcal{M}$. In the context of adversarial attacks, adversaries aim to maximize a certain objective function $\ell(\cdot)$ to mislead a target model $h(X; \boldsymbol{\theta})$ by searching for the worst perturbation for each instance (Goodfellow et al., 2015; Carlini & Wagner, 2017; Dong et al., 2018; Madry et al., 2018), where $\boldsymbol{\theta}$ stands for parameters of the target model. To formalize the intuition of adversarial

---

[1]These three causes are not indispensable. Deleting $\boldsymbol{\theta}$ gives black-box attacks (Papernot et al., 2017; Dong et al., 2018) that only use $(X, Y)$ to perform attacks. Deleting $X$ leads to universal adversarial attacks (Moosavi-Dezfooli et al., 2017; Hendrik Metzen et al., 2017), which assume that one adversarial perturbation is sufficient to fool DNNs. Recent works show that a specific transform to $X$ effectively evaluates the sensitivity of DNNs within a small neighborhood (Rahaman et al., 2019; Zhang et al., 2020b), which corresponds to deleting $(Y, \boldsymbol{\theta})$ from the cause set.

attacks, we can first search for the adversarial perturbation $E_{adv}$, and then use a function to approximate the mechanism for generating $E_{adv}$. The adversarial perturbation $E_{adv}$ can be obtained by maximizing $\ell(\cdot)$:

$$E_{adv} = \arg\max_{E' \in \mathbb{B}} \ell(h(X + E'; \boldsymbol{\theta}), Y). \tag{2}$$

where $\mathbb{B}$ is a set of valid perturbations, and the adversarial perturbation $E_{adv}$ is the result of the mechanism $\mathcal{M}_{adv}$, i.e., $E_{adv} \coloneqq \mathcal{M}_{adv}(X, Y, \boldsymbol{\theta}, U_E)$. The intervention distribution that corresponds to the adversarial mechanism $M_{adv}$ is defined as the adversarial distribution $P_{\boldsymbol{\theta}}(\tilde{X}, Y)$, where the subscript $\boldsymbol{\theta}$ indicates that $P_{\boldsymbol{\theta}}(\tilde{X}, Y)$ is crafted to attack the target model $h(X; \boldsymbol{\theta})$ with parameter $\boldsymbol{\theta}$.

## 3 METHOD

To understand the adversarial vulnerability, we study the difference between the natural and adversarial distributions and conclude that conditional associations between labels and style factors play a crucial role in adversarial vulnerability. Inspired by the conclusion, we propose a method to eliminate the difference in such conditional associations to mitigate adversarial vulnerability.

### 3.1 ORIGIN OF ADVERSARIAL VULNERABILITY

Inspired by the causal graph, we derive a causal understanding of adversarial vulnerability. According to the causal graph $\mathcal{G}$ depicted in Fig. 1, there is a path, $S \to \underline{X} \leftarrow C \to Y$, from the style variable $S$ to the label $Y$ when $X$ is given [2], which leads to the correlation between labels and style variables. The spurious correlation implies that DNNs can perform well on the training set by fitting the statistical association between $Y$ and $S$, even though the genuine content information is dropped. Moreover, if the conditional association between $Y$ and $S$ on the (natural) test set is similar to that on the training set, fitting these spurious correlations will also perform well on the test set. This is consistent with the recent work (Ilyas et al., 2019), which shows that training DNNs with incorrectly labeled data, i.e., the genuine content information is *not* utilized for training, yields good accuracy on the (natural) test set. However, suppose DNNs learn such spurious correlation. In that case, their performance will change with the conditional association between $Y$ and $S$. Consequently, fitting such spurious correlation gives adversaries a chance to fool DNNs. Therefore, the conditional association between labels and style variables is a key to understand adversarial vulnerability.

To identify the origin of adversarial vulnerability from the spurious correlation perspective, we study the distinction between the adversarial distribution and the natural distribution. To look at what makes the adversarial distribution different from the natural distribution, we expand the natural $P(Y|X)$ and the adversarial distribution $P_{\boldsymbol{\theta}}(Y|\tilde{X})$

$$P(Y|X) = \sum_{s \in \mathbb{S}} P(s|X) P(Y|X, s), \; P_{\boldsymbol{\theta}}(Y|\tilde{X}) = \sum_{s \in \mathbb{S}} P_{\boldsymbol{\theta}}(s|\tilde{X}) P_{\boldsymbol{\theta}}(Y|\tilde{X}, s), \tag{3}$$

where we assume the set of valid styles $\mathbb{S}$ is discrete, and $s$ stands for a certain style. We can see that the difference between $P(Y|X)$ and $P_{\boldsymbol{\theta}}(Y|\tilde{X})$ results from two terms, i.e., $P_{\boldsymbol{\theta}}(s|\tilde{X})$ and $P_{\boldsymbol{\theta}}(Y|\tilde{X}, s)$. $P_{\boldsymbol{\theta}}(s|\tilde{X})$ represents the change of style information, e.g., textures (Geirhos et al., 2018), transformations (Chen et al., 2020; Mitrovic et al., 2020; He et al., 2020), and domain shifts (Ganin et al., 2016). It is shown that changing image textures has a significant impact on predictions of DNNs (Geirhos et al., 2018), but such drastic style changes will hardly appear in the context of adversarial attacks as perturbations are required to be imperceptible [3]. Although modifying style variables $S$ is not allowed, adversaries can exploit the conditional association between $Y$ and $S$, i.e., $P_{\boldsymbol{\theta}}(Y|\tilde{X}, s)$, via injecting a specific perturbation to generate adversarial distributions. Specifically, under an appropriate soft intervention, i.e., $\mathcal{M}_{adv}$, the conditional distribution $P_{\boldsymbol{\theta}}(Y|\tilde{X}, s)$ can be drastically different from $P(Y|X, s)$. Namely, the property of adversarial distribution essentially results from the drastic difference in the statistical association between labels and style variables. Intuitively, if DNNs fit the conditional associations between $Y$ and $S$, their performance will change

---

[2]Here, we use the underlined variable $\underline{X}$ to represent that the variable $X$ is given.

[3]In this work, we focus on imperceptible adversarial examples, i.e., the widely studied $\ell_p$-norm bounded adversarial examples, where drastic style changes hardly appear.

with the spurious correlation. Consequently, a drastic difference between the conditional association between $Y$ and $S$ will cause significant performance degradation of DNNs.

Hence, the origin of adversarial vulnerability is the excessive focus of DNNs on spurious correlations between labels and style variables. This conclusion provides a causal perspective for the empirical observation, i.e. some features are useful but not robust (Ilyas et al., 2019), so adversarial examples can be viewed as a model phenomenon rather than merely a human phenomenon.

## 3.2 THE ADVERSARIAL DISTRIBUTION ALIGNMENT METHOD

In light of the causal understanding of the adversarial vulnerability, improving adversarial robustness requires forcing DNNs to fit the causal relations rather than merely the statistical associations. However, only the perceived data can be observed in practice, and the supervised information of content variables is usually unavailable. Therefore, an approach that can avoid reliance on such supervised information is required.

To develop such an approach, we revisit the difference between the adversarial and natural distributions. Intuitively, if the difference between the natural and the adversarial distribution is negligible, the adversarial vulnerability of DNNs can be mitigated, as DNNs perform well on the natural distribution (Krizhevsky et al., 2012; He et al., 2016; Zhang et al., 2021). Consequently, a straightforward solution for improving adversarial robustness is to align these two distributions. The aforementioned analysis shows that the property of adversarial distributions to fool DNNs comes from specific conditional associations between labels $Y$ and style variables $S$, i.e., $P_{\theta}(Y|\tilde{X}, s)$ and $P(Y|X, s)$. Inspired by the conclusion, we propose an adversarial distribution alignment method to eliminate the difference between the natural and adversarial distributions. Specifically, we regard the natural distribution $P(Y|X, s)$ as an *anchor*, then align the adversarial distribution $P_{\theta}(Y|\tilde{X}, s)$ with the anchor such that the difference between these two distributions is negligible. In addition, the relationship between $Y$ and $X$ should also be the same on both the adversarial and natural distributions. Concretely, we operationalize the intuition of the adversarial distribution alignment by:

$$\min_{\theta} d\left(P\left(Y|X\right), P_{\theta}\left(Y|\tilde{X}\right)\right) + \lambda \mathbb{E}_s d\left(P\left(Y|X, s\right), P_{\theta}\left(Y|\tilde{X}, s\right)\right). \tag{4}$$

Here $d(\cdot)$ is a metric reflecting the divergence between two distributions, and $\lambda > 0$ presents the weighting of the misalignment penalty. In addition, we assume $X$ and $\tilde{X}$ have the same support set because adversarial perturbations are usually bounded. Solving the adversarial distribution alignment objective is equivalent to search a classifier $h(\tilde{X}; \theta)$ such that the adversarial distribution of the classifier is similar to the natural distribution. Benefiting from the consistency of these two distributions, classifier $h(\tilde{X}; \theta)$ can perform well on both the natural and adversarial distributions.

In practice, the adversarial distribution often cannot be obtained analytically, so we relax the distribution divergence in Eq. 4 to the sum of two divergences, see Appendix A for details. For the divergence between $P(Y|X)$ and $P_{\theta}(Y|\tilde{X})$, we have:

$$KL\left(P_{\theta}\left(Y|\tilde{X}\right), Q_{\theta}\left(Y|\tilde{X}\right)\right) + \gamma KL\left(P\left(Y|X\right), Q_{\theta}\left(Y|X\right)\right)$$
$$= \mathbb{E}_{(\tilde{X}, Y) \sim P_{\theta}(\tilde{X}, Y)} CE\left(h\left(\tilde{X}; \theta\right), Y\right) + \gamma \mathbb{E}_{(X, Y) \sim P(X, Y)} CE\left(h\left(X; \theta\right), Y\right)$$
$$\approx \mathbb{E}_{(X, Y) \sim P(X, Y)} CE\left(h\left(X + E_{adv}; \theta\right), Y\right) + \gamma CE\left(h\left(X; \theta\right), Y\right), \tag{5}$$

where $KL$ is the Kullback-Leibler divergence, $CE$ is the cross-entropy loss, $Q_{\theta}(Y|X)$ is the conditional distribution specified by the classifier $h(X; \theta)$, and $\gamma$ is a tunable hyperparameter. Because adversarial examples are usually generated by adding perturbation to their corresponding natural samples rather than sampled from the adversarial distribution independently, we use $X + E_{adv}$ to approximate examples independently sampled from the adversarial distribution, where $X$ is sampled from the natural distribution independently.

Similar to Eq. 5, the divergence between $P(Y|\tilde{X}, s)$ and $P_{\theta}(Y|\tilde{X}, s)$ can be approximated by:

$$\mathbb{E}_{(X, Y) \sim P(X, Y)} CE\left(g\left(s\left(X + E_{adv}\right); W_g\right), Y\right) + \beta\left(g\left(s\left(X\right); W_g\right), Y\right), \tag{6}$$

where $g\left(s\left(X\right); W_g\right)$ is a function used for modeling the statistically conditional association between labels and style variables, $s\left(X\right)$ stands for the integrated representation of $X$ and $s$, and $\beta$

is a tunable hyperparameter. Thus, the overall objective of the proposed adversarial distribution alignment method can be expressed as

$$
\min_{\boldsymbol{\theta}, W_g} \mathbb{E}_{(X,Y) \sim P(X,Y)} CE\left(h\left(X + E_{adv}; \boldsymbol{\theta}\right), Y\right) + \gamma CE\left(h\left(X; \boldsymbol{\theta}\right), Y\right)
$$
$$
+ \lambda\left(\mathbb{E}_s CE\left(g\left(s\left(X + E_{adv}\right); W_g\right), Y\right) + \beta CE\left(g\left(s\left(X\right); W_g\right), Y\right)\right). \tag{7}
$$

To make sure that introducing $g$ can benefit learning $h$, it is necessary to design an approach to connect model $g$ and model $h$. In this paper, we connect $g$ and $h$ by representation sharing. Interestingly, according to Eq. 4 and Eq. 7, if we omit the spurious correlation between labels and style variables, i.e., $\lambda = 0$, and set $\gamma = 0$, Eq. 7 then becomes the objective function introduced by Madry (Madry et al., 2018). This suggests that the proposed adversarial distribution alignment method is consistent with the seminal variant Madry (Madry et al., 2018) of adversarial training.

### 3.3 REALIZATION OF ADVERSARIAL DISTRIBUTION ALIGNMENT METHOD

According to Eq. 7, realizing the proposed adversarial distribution alignment method requires predicting labels with all style variables, i.e., modeling the statistically conditional association between $Y$ and $S$. However, there are two major obstacles preventing us from calculating the last term of Eq. 7. Specifically, a) the number of all possible styles $s$ is infinite, so the cost of calculating the expectation in Eq. 7 can grow to infinity; b) the representation of the integrated representation of $X$ and $s$, i.e., $s\left(X\right)$, used for predicting labels is unknown.

To calculate the expectation in Eq. 7, we have to make some assumptions to approximate the distribution of the style variable, as the true distribution is unknown. Following previous work (Gal & Ghahramani, 2016; Kendall & Gal, 2017), we assume a Gaussian distribution to approximate the unknown distribution. Specifically, the representation $s\left(X\right)$ is estimated by $\hat{s}\left(X\right)$ sampled from a Gaussian distribution, i.e., $\hat{s}\left(X\right) \sim \mathcal{N}\left(\mu\left(X\right), \Sigma\right)$, where $\mu\left(X\right)$ is employed to model the Gaussian distribution's mean, and $\Sigma$ is the covariance matrix. Because little prior knowledge about the covariance matrix is observed, we simply regard the covariance matrix $\Sigma$ as an identity matrix, i.e., $\hat{s}\left(X\right) \sim \mathcal{N}\left(\mu\left(X\right), \sigma^2 I\right)$. Then, the expectation can be approximated by

$$
\mathbb{E}_s CE\left(g\left(s\left(X\right); W_g\right), Y\right) \approx \mathbb{E}_{\hat{s}(X) \sim \mathcal{N}(\mu(X), \sigma^2 I)} CE\left(g\left(\hat{s}\left(X\right); W_g\right), Y\right). \tag{8}
$$

Thanks to the Gaussian distribution assumption, we can derive an upper bound of the expectation:

$$
\mathbb{E}_{\hat{s}(X) \sim \mathcal{N}(\mu(X), \sigma^2 I)} CE\left(g\left(\hat{s}\left(X\right); W_g\right), Y\right) \leq CE\left(\overline{g}\left(\mu\left(X\right); W_g\right), Y\right), \tag{9}
$$

where the probability of the $i^{th}$ category of $\overline{g}\left(\mu\left(X\right); W_g\right)$ is (see Appendix B for details)

$$
P\left(Y = i | \overline{g}\left(\mu\left(X\right); W_g\right)\right) = \frac{e^{W_{g,i}^\top \mu(X)}}{\sum_j e^{W_{g,j}^\top \mu(X) + \frac{\sigma^2}{2}(W_{g,j} - W_{g,i})^\top (W_{g,j} - W_{g,i})}}. \tag{10}
$$

Here, we simply set $g$ to a linear function, i.e., $W_g$ is a linear mapping. Instead of calculating all styles $\hat{s}\left(X\right)$, the conclusion of Eq. 9 and Eq. 10 shows that only the mean and variance of $\hat{s}\left(X\right)$ are required to calculate the expectation in Eq. 7.

In light of Eq. 9 and Eq. 10, the challenge of learning the representation of $s\left(X\right)$ boils down to estimating the mean of $\hat{s}\left(X\right)$, i.e., no need to estimate every style representation explicitly. In addition, the variance can be treated as a hyperparameter. To estimate the mean, we draw inspiration from the proposed causal graph. According to the causal graph $\mathcal{G}$, the content and style variables are statistically independent, i.e., $C \perp\!\!\!\perp S$. Thus, the estimated content $\hat{c}\left(X\right)$ and style $\hat{s}\left(X\right)$ are desired to be independent, i.e., $\hat{c}\left(X\right) \perp\!\!\!\perp \hat{s}\left(X\right)$. The underlying intuition is that we aim to model the causal relation between the content variable and the style variable and omit the spurious correlation between $\hat{c}\left(X\right)$ and $\hat{s}\left(X\right)$, because the spurious correlation is not a causal relation, although $\hat{c}\left(X\right)$ and $\hat{s}\left(X\right)$ are not statistically independent. Following previous work (LeCun et al., 1998; Bengio et al., 2013), we regard DNNs as a combination of representation learning modules and linear classifier modules. Specifically, we apply a linear function to the learned representation for approximating the content which is further used to predicting labels. Specifically, we have $\hat{c}\left(X\right) = \hat{c}\left(r\left(X; W_r\right); W_c\right) = W_c r\left(X; W_r\right)$. That is, $W_c$ are parameters used for predicting labels. According to the Gaussian distribution assumption, we can obtain the estimated style by the reparameterization trick, i.e., $\hat{s}\left(X\right) = \mu\left(X; W_s\right) + \sigma \boldsymbol{n}$, where $W_s$ presents

parameters for modeling the mean, and $n$ is sampled from a normal distribution. For simplicity, we assume that $\mu(X; W_s)$ is an affine mapping applied to the learned representation [4], i.e., $\mu(X; W_s) = \mu(r(X; W_r); W_s) = W_s r(X; W_r)$. Assume that $r(X; W_r)$ is a Gaussian distribution with a covariance matrix $M$. Then, the independence $\hat{c}(X) \perp\!\!\!\perp \hat{s}(X)$ holds if we set $W_s$ as an instantiation of the orthogonal complement of $W_c$, i.e., $\ker(W_c)^\perp \perp_M \ker(W_s)^\perp$. Here, we define $\langle a, b \rangle_M = \langle a, Mb \rangle$, and the orthogonality of two subspaces is defined likewise. Therefore, we can simply employ the learned representation $r(X; W_r)$ and the parameters used for predicting labels, i.e., $W_c$, to estimate $\mu(X)$, see Appendix C for details.

Combining Eq. 7 and Eq. 9, we derive a realization of adversarial distribution alignment method:

$$\min_{\boldsymbol{\theta}, W_g} \mathbb{E}_{(X,Y) \sim P(X,Y)} CE\left(h\left(X + E_{adv}; \boldsymbol{\theta}\right), Y\right) + \gamma CE\left(h\left(X; \boldsymbol{\theta}\right), Y\right)$$
$$+ \lambda\left(CE\left(\overline{g}\left(\mu\left(X + E_{adv}\right); W_g\right), Y\right) + \beta CE\left(\overline{g}\left(\mu\left(X\right); W_g\right), Y\right)\right). \tag{11}$$

Given $X$, Eq. 11 encourages the statistically conditional association between labels and style variables of the adversarial distribution to be close to that of the natural distribution. The explicit distribution alignment can reduce the difference between the adversarial distribution and the natural distribution. If the conditional association of the adversarial distribution is similar to that of the natural distribution, it should be hard for the adversary to find adversarial examples, which is consistent with recent work (Kilbertus et al., 2018; Schölkopf et al., 2021).

## 4 EXPERIMENT

### 4.1 SETUPS

**Baseline methods.** Our experiments are designed to demonstrate the necessity of considering the spurious correlation between labels $Y$ and style variables $S$ when developing robust models. Eq. 7 shows that if we set the hyperparameter $\gamma = 0$, and omit the spurious correlation between $Y$ and $S$, the proposed method is equivalent to the adversarial training variant Madry (Madry et al., 2018). Thus, to demonstrate the necessity of considering the spurious correlation, we set $\gamma = 0$ and $\lambda > 0$ in Eq. 7, named CausalAdv-M, and compare the adversarial robustness of CausalAdv-M with that of Madry. In addition, the model capacity is often insufficient in adversarial training (Madry et al., 2018), so replacing one-hot labels of the first term in Eq. 7 with soft targets, i.e., the model prediction $h(X; \boldsymbol{\theta})$, can relieve the problem of insufficient model capacity [5]. Considering the insufficient model capacity, we replace $Y$ in the first term of Eq. 7 with the model prediction, and the derived method is called CausalAdv-T. We find that CausalAdv-T becomes the objective introduced in TRADES (Zhang et al., 2019) when we omit the spurious correlation, i.e., $\lambda = 0$, which shows that the proposed method also shares the same spirits to TRADES. Therefore, to demonstrate the importance of the spurious correlation between $Y$ and $S$, we compare CausalAdv-M and CausalAdv-T with Madry and TRADES, respectively.

**Evaluation metrics and training details.** To evaluate the robustness for different methods, we compute the test accuracy on natural and adversarial examples with $\ell_\infty$-norm bounded perturbation generated by: FGSM (Goodfellow et al., 2015), PGD (Madry et al., 2018), and C&W (Carlini & Wagner, 2017) attacks. The robustness is evaluated on both the best checkpoint model suggested by (Rice et al., 2020) and the last checkpoint model used in (Madry et al., 2018), respectively. For MNIST, we use the same CNN architecture as (Carlini & Wagner, 2017; Zhang et al., 2019). For CIFAR10 and CIFAR100, two architectures are employed: ResNet-18 (He et al., 2016) and WRN-34-10 (Zagoruyko & Komodakis, 2016). The settings of attacks and hyper-parameters for training are the same as previous works, more details can be found in Appendix D.

### 4.2 ROBUSTNESS EVALUATION

We evaluate the robustness of Madry, TRADES, and the proposed method on MNIST, CIFAR10, and CIFAR100 against various attacks (Goodfellow et al., 2015; Madry et al., 2018; Carlini & Wagner, 2017), which are widely used in the literature. We report the classification accuracy on MNIST in

---

[4] Studying non-linear functions is an interesting open question, and we leave it as future work.

[5] Knowledge distillation (Hinton et al., 2015) shows that models with insufficient capacity prefer soft targets.

Table 1: Classification accuracy (%) on MNIST under the white-box threat model with $\epsilon = 0.3$. The best-performance model and the corresponding accuracy are highlighted.

| Method | Best checkpoint | | | | Last checkpoint | | | |
|---|---|---|---|---|---|---|---|---|
| | Natural | FGSM | PGD-40 | CW-40 | Natural | FGSM | PGD-40 | CW-40 |
| Madry | 99.48 | 97.82 | 95.75 | 95.92 | 99.47 | 96.52 | 94.33 | 94.45 |
| CausalAdv-M | $\mathbf{99.53}_{\pm 0.04}$ | $\mathbf{98.02}_{\pm 0.07}$ | $\mathbf{96.37}_{\pm 0.12}$ | $\mathbf{96.47}_{\pm 0.17}$ | $\mathbf{99.49}_{\pm 0.08}$ | $\mathbf{96.83}_{\pm 0.10}$ | $\mathbf{94.67}_{\pm 0.14}$ | $\mathbf{94.84}_{\pm 0.19}$ |
| TRADES | 99.39 | 97.22 | 96.55 | 96.66 | 99.36 | 96.76 | 94.89 | 94.91 |
| CausalAdv-T | $\mathbf{99.49}_{\pm 0.06}$ | $\mathbf{97.82}_{\pm 0.07}$ | $\mathbf{96.72}_{\pm 0.10}$ | $\mathbf{96.78}_{\pm 0.15}$ | $\mathbf{99.49}_{\pm 0.04}$ | $\mathbf{97.32}_{\pm 0.08}$ | $\mathbf{96.63}_{\pm 0.13}$ | $\mathbf{96.69}_{\pm 0.21}$ |

Table 2: Classification accuracy (%) of ResNet-18 on CIFAR-10 under the white-box threat model with $\epsilon = 8/255$. The best-performance model and the corresponding accuracy are highlighted.

| Method | Best checkpoint | | | | Last checkpoint | | | |
|---|---|---|---|---|---|---|---|---|
| | Natural | FGSM | PGD-20 | CW-20 | Natural | FGSM | PGD-20 | CW-20 |
| Madry | **83.56** | 56.69 | 51.92 | 51.00 | **84.65** | 54.37 | 46.38 | 46.73 |
| CausalAdv-M | $80.42_{\pm 0.39}$ | $\mathbf{57.98}_{\pm 0.21}$ | $\mathbf{54.44}_{\pm 0.18}$ | $\mathbf{52.51}_{\pm 0.25}$ | $83.72_{\pm 0.41}$ | $\mathbf{59.17}_{\pm 0.24}$ | $\mathbf{51.82}_{\pm 0.19}$ | $\mathbf{50.93}_{\pm 0.27}$ |
| TRADES | **81.39** | 57.25 | 53.64 | 51.39 | **82.91** | 57.95 | 52.80 | 51.27 |
| CausalAdv-T | $81.22_{\pm 0.27}$ | $\mathbf{58.97}_{\pm 0.17}$ | $\mathbf{54.55}_{\pm 0.16}$ | $\mathbf{52.95}_{\pm 0.26}$ | $81.62_{\pm 0.30}$ | $\mathbf{58.90}_{\pm 0.16}$ | $\mathbf{53.64}_{\pm 0.14}$ | $\mathbf{52.70}_{\pm 0.37}$ |

Table 1, where "Natural" denotes the accuracy on natural test images. We denote by PGD-40 the PGD attack with 40 iterations for generating adversarial examples, which also applies to the C&W attack. The results of ResNet-18 on CIFAR10 and CIFAR100 are illustrated in Table 2 and Table 3, respectively. The results of WRN-34-10 are in Appendix E. We can see that the proposed method achieves the best robustness against all three types of attacks, demonstrating that taking into account the spurious correlation can significantly improve the adversarial robustness. To further understand the comparative effects of different terms of the proposed method, we reorganize the robust accuracy of the best checkpoint trained on CIFAR-10 and CIFAR-100 in Appendix F.

## 4.3 DISCUSSION

**Consideration of gradient obfuscation.** According to the criterion suggested by (Athalye et al., 2018), we exclude the potential effect of gradient obfuscation by showing the following phenomenons: a) the performance of our method on FSGM attack (59.17%) is better than iterative attacks PGD-20 (51.82%) and C&W-20 (50.93%); b) the performance of our method on black-box PGD-20 (83.65%) and C&W-20 (83.57%) attacks is better than that on white-box attacks (51.82%); c) strong attacks cause lower accuracy than weak attacks, i.e., the accuracy on PGD-20 and PGD-100 are 51.82% and 48.05%, respectively. In addition, no gradient shattering operator is used in our method. All these results are evaluated on the CIFAR10 dataset using the last checkpoint of ResNet-18. Thus, according to the criterion suggested by (Athalye et al., 2018), the robustness improvement of the proposed method does *not* result from gradient obfuscation.

**Consideration of adaptive attack.** According to the adaptive attack criterion (Tramer et al., 2020),

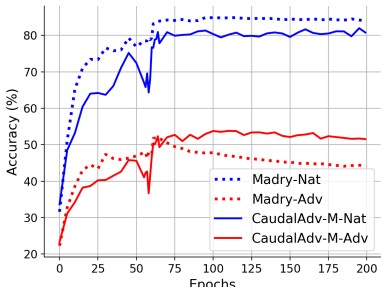

Figure 2: Comparisons of Madry (dotted lines) and CausalAdv-M (solid lines) using ResNet-18 on the CIFAR-10 dataset under PGD-20 attack. Madry-Nat and Madry-Adv represent the accuracy of Madry on the natural and adversarial data, respectively, which also applies to CausalAdv-M. To verify that CausalAdv-M effectively alleviates the robust overfitting rather than delaying the occurrence of robust overfitting, we train these models with 200 epochs which is larger than the basic setting.

we augment the original objective function used in the PGD attack with the proposed adversarial distribution alignment loss to implement adaptive attacks, i.e., Eq. 11. Under the adaptive attack, the accuracy is 51.68%, while under the original PGD-20 attack is 51.82%, demonstrating that the proposed method is genuinely robust.

**Consideration of AutoAttack.** Following previous work (Pang et al., 2020), we verify the effectiveness of the proposed method on AutoAttack (Croce & Hein, 2020). According to the commonly used setting, see (Pang et al., 2020), we report robust accuracy of WRN-34-10 trained with CIFAR10 dataset on Auto-Attack, Madry: 49.58%, CausalAdv-M: 51.56%, TRADES: 52.46%, CausalAdv-

Table 3: Classification accuracy (%) of ResNet-18 on CIFAR-100 under the white-box threat model with $\epsilon = 8/255$. The best-performance model and the corresponding accuracy are highlighted.

| Method | Best checkpoint | | | | Last checkpoint | | | |
|---|---|---|---|---|---|---|---|---|
| | Natural | FGSM | PGD-20 | CW-20 | Natural | FGSM | PGD-20 | CW-20 |
| Madry | **55.98** | 28.39 | 25.15 | 24.04 | **55.08** | 25.35 | 21.63 | 21.42 |
| CausalAdv-M | $54.07_{\pm 0.17}$ | $\mathbf{29.76}_{\pm 0.16}$ | $\mathbf{27.62}_{\pm 0.13}$ | $\mathbf{25.44}_{\pm 0.15}$ | $54.81_{\pm 0.23}$ | $\mathbf{26.83}_{\pm 0.19}$ | $\mathbf{23.34}_{\pm 0.15}$ | $\mathbf{22.93}_{\pm 0.13}$ |
| TRADES | **53.85** | 29.04 | 27.91 | 24.09 | 53.54 | 29.29 | 26.80 | 23.79 |
| CausalAdv-T | $53.17_{\pm 0.39}$ | $\mathbf{30.66}_{\pm 0.20}$ | $\mathbf{28.57}_{\pm 0.18}$ | $\mathbf{25.74}_{\pm 0.18}$ | $\mathbf{54.79}_{\pm 0.41}$ | $\mathbf{30.81}_{\pm 0.40}$ | $\mathbf{28.51}_{\pm 0.35}$ | $\mathbf{25.32}_{\pm 0.27}$ |

T: 54.09%, and HE (Pang et al., 2020): 53.74%, where HE is an adversarial variant achieving state-of-the-art performance. These results show that the proposed method can endow models with robustness comparable to the state-of-the-art performance.

**Mitigating robust overfitting.** The recent work (Rice et al., 2020) first studies the robust overfitting phenomenon. Robust overfitting means that further training will increase the robust training accuracy and test accuracy on natural data after a certain training epoch, but the robust test accuracy will decrease. The robust overfitting phenomenon of Madry (Madry et al., 2018) is depicted in Fig. 2. It can be seen that the robust test accuracy of Madry decreases to about 44%, while the best robust accuracy of Madry is 51.92%. In contrast, the proposed method drastically reduces the difference between the best robust accuracy 54.44% and the robust accuracy 50.93% of the last checkpoint.

## 5 RELATED WORK

**Adversarial attack.** Unlike the assumption employed in noisy labels (Han et al., 2020; Liu & Tao, 2015; Xia et al., 2020), adversarial attacks assume that the input determines noise. Since the realization of the adversarial example phenomenon (Biggio et al., 2013; Szegedy et al., 2014), tons of adversarial attacks have been proposed (Moosavi-Dezfooli et al., 2016; Goodfellow et al., 2015; Carlini & Wagner, 2017; Dong et al., 2018; Tu et al., 2019; Madry et al., 2018; Croce & Hein, 2020). Among these attacks, FGSM Goodfellow et al. (2015), PGD attack (Madry et al., 2018), C&W attack (Carlini & Wagner, 2017), and Auto-Attack (Croce & Hein, 2020) are the most commonly used attacks for evaluating robustness.

**Adversarial defense.** The development of adversarial attacks promotes the progress of adversarial defense and detection. Recent work on improving adversarial robustness mainly falls into two categories: certified defense (Raghunathan et al., 2018; Wong & Kolter, 2018; Singla & Feizi, 2020) and empirical defense and detection with two-sample test (Najafi et al., 2019; Carmon et al., 2019; Shafahi et al., 2019; Wong et al., 2019; Pang et al., 2020; Rice et al., 2020; Ma et al., 2018; Gao et al., 2021). Detailed discussions of these exciting works can be found in Appendix G.

**Causal reasoning.** The field of graphical causality, like machine learning, has a long history, see Pearl (2009); Schölkopf et al. (2021); Peters et al. (2017). One purpose of causal reasoning is to pursue the causal effect of interventions, contributing to achieving the desired objectives. Recent work shows the benefits of introducing causality into machine learning from various aspects (Zhang et al., 2020a; Mitrovic et al., 2020; Teshima et al., 2020; Tang et al., 2020; Sauer & Geiger, 2020; Tang et al., 2021). More details about relevant works can be found in Appendix H.

## 6 CONCLUSION

In this paper, we provide a novel causality viewpoint for understanding and mitigating adversarial vulnerability. Through constructing the causal graph of the adversarial data generation process and formalizing the intuition of adversarial attacks, we show that the spurious correlation between labels and style variables is important for understanding and mitigating adversarial vulnerability. Inspired by the observation, we propose the adversarial distribution alignment method, which takes the spurious correlation into account for robustness improvement. In addition, we find that the proposed method shares the same spirits to existing adversarial training variants. In future work, we will develop more effective algorithms to leverage or eliminate the spurious correlation between labels and style variables to further improve adversarial robustness. In addition, we will explore the uses of counterfactual statements to explain and mitigate the adversarial vulnerability. In sum, we make a first step towards employing causality to contribute to adversarial learning.

## 7 ACKNOWLEDGMENT

We thank the area chair and reviewers for their valuable comments. YGZ and BH were supported by the RGC ECS No. 22200720 and NSFC YSF No. 62006202. YGZ and XMT were supported by NSFC No. 61872329. MMG was supported by Australian Research Council Project DE210101624. TLL was supported by Australian Research Council Projects DE-190101473 and DP-220102121. KZ would like to acknowledge the support by the National Institutes of Health (NIH) under Contract R01HL159805, by the NSF-Convergence Accelerator Track-D award #2134901, and by the United States Air Force under Contract No. FA8650-17-C7715.

## 8 ETHICS STATEMENT

This paper does not raise any ethics concerns. This study does not involve any human subjects, practices to data set releases, potentially harmful insights, methodologies and applications, potential conflicts of interest and sponsorship, discrimination/bias/fairness concerns, privacy and security issues, legal compliance, and research integrity issues.

## 9 REPRODUCIBILITY STATEMENT

To ensure the reproducibility of experimental results, we open source our code https://github.com/YonggangZhangUSTC/CausalAdv.git.

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

## A  DERIVATION OF DISTRIBUTION DIVERGENCE

Considering that the adversarial distribution often cannot be obtained analytically, we need to relax the distribution divergence in Eq. 4, i.e., $d\left(P\left(Y|X\right), P_{\boldsymbol{\theta}}\left(Y|\tilde{X}\right)\right)$. Without loss of generality, the metric $d$ in Eq. 4 can be realized as total variation distance (TVD). Thus, according to the definition of TVD, we have

$$d\left(P\left(Y|X\right), P_{\boldsymbol{\theta}}\left(Y|\tilde{X}\right)\right) \leq d\left(P_{\boldsymbol{\theta}}\left(Y|\tilde{X}\right), Q_{\boldsymbol{\theta}}\left(Y|\tilde{X}\right)\right) + d\left(P\left(Y|X\right), Q_{\boldsymbol{\theta}}\left(Y|X\right)\right), \quad (12)$$

where $Q_{\boldsymbol{\theta}}(Y|X)$ is the conditional distribution specified by the classifier $h\left(X;\boldsymbol{\theta}\right)$. According to the Pinsker inequality, i.e., $d\left(P, Q\right) \leq \sqrt{\frac{KL(P||Q)}{2}}$, where $KL$ is the Kullback-Leibler divergence, we have an upper bound of Eq. 12:

$$d\left(P\left(Y|X\right), P_{\boldsymbol{\theta}}\left(Y|\tilde{X}\right)\right) \leq \sqrt{\frac{KL\left(P_{\boldsymbol{\theta}}\left(Y|\tilde{X}\right)||Q_{\boldsymbol{\theta}}\left(Y|\tilde{X}\right)\right)}{2}} + \sqrt{\frac{KL\left(P\left(Y|X\right)||Q_{\boldsymbol{\theta}}\left(Y|X\right)\right)}{2}}. \tag{13}$$

Now, we can employ the upper bound Eq. 13 as a surrogate loss. In practice, we can replace $\sqrt{\frac{KL(P||Q)}{2}}$ with $KL\left(P||Q\right)$. The intuition is that the difference between $\sqrt{\frac{KL(P||Q)}{2}}$ and $KL\left(P||Q\right)$ is relatively small when $KL\left(P||Q\right)$ is not extremely large, so the replacement will not introduce much difference. Thus, we derive Eq. 5: $KL\left(P_{\boldsymbol{\theta}}\left(Y|\tilde{X}\right)||Q_{\boldsymbol{\theta}}\left(Y|\tilde{X}\right)\right) + \gamma KL\left(P\left(Y|X\right)||Q_{\boldsymbol{\theta}}\left(Y|X\right)\right)$, where $\gamma$ is a tunable hyperparameter.

Here, we give an intuitive explanation of why optimizing Eq. 5 can make the adversarial distribution similar to the natural distribution. Considering that both the adversarial distribution $P_{\theta}\left(Y|\tilde{X}\right)$ and the anchor $Q_{\theta}\left(Y|\tilde{X}\right)$ are parameterized by $\theta$, so optimizing $\theta$ to minimize $KL\left(P_{\theta}\left(Y|\tilde{X}\right)||Q_{\theta}\left(Y|\tilde{X}\right)\right)$ can force the adversarial distribution be similar to the distribution specified by the classifier $h\left(X;\theta\right)$, i.e., $Q_{\theta}\left(Y|\tilde{X}\right)$. That is, optimizing $Q_{\theta}\left(Y|\tilde{X}\right)$ will also change the adversarial distribution $P_{\theta}\left(Y|\tilde{X}\right)$. In addition, minimizing the divergence $KL\left(P\left(Y|X\right)||Q_{\theta}\left(Y|\tilde{X}\right)\right)$ can endow the classifier with the ability to provide a good prediction performance. Thus, minimizing $KL\left(P_{\boldsymbol{\theta}}\left(Y|\tilde{X}\right)||Q_{\boldsymbol{\theta}}\left(Y|\tilde{X}\right)\right)$ will make the adversarial distribution be similar with the natural distribution.

To verify the replacement will not introduce much difference, we evaluate the robustness of CausalAdv-M and CausalAdv-T on the CIFAR-10 dataset with two losses, i.e., Eq. 5 and Eq. 13. The results evaluated on best checkpoints are shown in Table 4.

Table 4: Classification accuracy (%) on CIFAR-10 under the white-box threat model with $\epsilon = 8/255$. Here, "*" means that Eq. 13 is used for optimization.

|  | Natural | FGSM | PGD20 | CW20 |
|---|---|---|---|---|
| Madry | 83.56 | 56.69 | 51.92 | 51.00 |
| CausalAdv-M | 80.42 | 57.98 | 54.44 | 52.51 |
| CausalAdv-M* | 80.03 | 57.47 | 52.98 | 52.72 |
| TRADES | 81.39 | 57.25 | 53.64 | 51.39 |
| CausalAdv-T | 81.22 | 58.97 | 54.55 | 52.95 |
| CausalAdv-T* | 79.94 | 58.46 | 54.12 | 52.28 |

## B  CALCULATION OF THE EXPECTATION ON THE STYLE INFORMATION

We provide details of calculating $\mathbb{E}_{\hat{s}\left(\tilde{X}\right) \sim \mathcal{N}\left(\mu\left(\tilde{X}\right), \sigma^2 I\right)} CE\left(g\left(\hat{s}\left(\tilde{X}\right); W_g\right), Y\right)$. We assume a normal distribution for the styles, i.e., $\hat{s}\left(\tilde{X}\right) \sim \mathcal{N}\left(\mu\left(\tilde{X}\right), \sigma^2 \boldsymbol{I}\right)$. According to the definition of the cross-entropy loss, for a input pair $(x, y)$ we have:

$$\mathbb{E}_{\hat{s}(x) \sim \mathcal{N}(\mu(x), \sigma^2 I)} CE\left(g\left(\hat{s}(x); W_g\right), y\right)$$

$$= \mathbb{E}_{\hat{s}(x) \sim \mathcal{N}(\mu(x), \sigma^2 I)} \log \frac{1}{P(Y = y | g\left(\hat{s}(x); W_g\right)))}$$

$$\leq \log \frac{1}{\mathbb{E}_{\hat{s}(x) \sim \mathcal{N}(\mu(x), \sigma^2 I)} P(Y = y | g\left(\hat{s}(x); W_g\right)))}$$

$$= \log \frac{1}{\mathbb{E}_{\hat{s}(x) \sim \mathcal{N}(\mu(x), \sigma^2 I)} \frac{e^{W_{g,y}^\top \hat{s}(x)}}{\sum_j e^{W_{g,j}^\top \hat{s}(x)}}}$$

$$= \log \mathbb{E}_{\hat{s}(x) \sim \mathcal{N}(\mu(x), \sigma^2 I)} \sum_j e^{(W_{g,j} - W_{g,y})^\top \hat{s}(x)} \tag{14}$$

$$= \log \sum_j e^{(W_{g,j} - W_{g,y})^\top \mu(x) + \frac{1}{2}(W_{g,j} - W_{g,y})^\top \sigma^2 I(W_{g,j} - W_{g,y})}$$

$$= \log \frac{\sum_j e^{W_{g,j}^\top \mu(x) + \frac{\sigma^2}{2}(W_{g,j} - W_{g,y})^\top (W_{g,j} - W_{g,y})}}{W_{g,y}^\top \mu(x)}$$

$$\triangleq \log \frac{1}{P\left((Y = y | \overline{g}\left(\mu(x); W_g\right)\right)}$$

$$\triangleq CE\left(\overline{g}\left(\mu(x); W_g\right), y\right),$$

where the inequality follows from the Jensen's inequality: $\mathbb{E}\log(X) \leq \log \mathbb{E}X$, the expectation is calculated by leveraging the moment-generating function:

$$\mathbb{E}e^{tX} = e^{t\mu + \frac{1}{2}\sigma^2 t^2}, X \sim \mathcal{N}(\mu, \sigma^2). \tag{15}$$

Note that, we define the function $\overline{g}\left(\mu(x); W_g\right)$ for simplicity:

$$P\left((Y = y | \overline{g}\left(\mu(x); W_g\right)\right) \triangleq \frac{W_{g,y}^\top \mu(x)}{\sum_{j=1} e^{W_{g,j}^\top \mu(x) + \frac{\sigma^2}{2}(W_{g,j} - W_{g,y})^\top (W_{g,j} - W_{g,y})}}. \tag{16}$$

Besides seting $g$ to a linear model, non-linear models, e.g., neural networks, can also be employed in practice. To verify the influence introduced by selecting different instantiation of model $g$, we compare different realizations of mode $g$, i.e., linear mapping and non-linear neural networks. The following results suggest that our method is relatively robust to the selection of model $g$. Note that, all these results are evaluated on the best checkpoint models trained on CIFAR-10 dataset.

Table 5: Classification accuracy (%) on CIFAR-10 under the white-box threat model with $\epsilon = 8/255$. Here, "+ Linear" means that model $g$ is instantiated with a linear mapping, and "+ Non-linear" means that model $g$ is instantiated with a neural network.

|  | Natural | FGSM | PGD20 | CW20 |
|---|---|---|---|---|
| Madry | 83.56 | 56.69 | 51.92 | 51.00 |
| CausalAdv-M + Linear | 80.68 | 57.18 | 53.36 | 51.41 |
| CausalAdv-M + Non-linear | 80.42 | 57.98 | 54.44 | 52.51 |
| TRADES | 81.39 | 57.25 | 53.64 | 51.39 |
| CausalAdv-T + Linear | 80.31 | 58.43 | 54.31 | 52.25 |
| CausalAdv-T + Non-linear | 81.22 | 58.97 | 54.55 | 52.95 |

## C  RELATIONSHIP BETWEEN ORTHOGONALITY AND STATISTICAL INDEPENDENCE

We give the proof for the following lemma in Sec. 3.3. Note that, we use $R$ to present the learned representation of $X$, and replace $X$ with $R$ for simplicity.

**Lemma 1.** $R \in \mathbb{R}^d$ *is the learned representation, where $d$ is the number of dimension of $R$. Assume that $R$ is a normal distribution with mean $m$ and covariance matrix $M$. The content used for predicting labels, i.e., logits, is obtained by applying a linear functions to $R$, i.e., $\hat{c}(R) = W_c R$, where $W_c$ are parameters used for mapping $R$ to logits. The style is modeled by a normal distribution, i.e., $\hat{s}(R) = \mu(R; W_s) + \Sigma(Y)^{\frac{1}{2}} \boldsymbol{n}$, where $W_s$ presents parameters for modeling the mean of styles, and $\boldsymbol{n}$ is sampled from a standard normal distribution. Assume that $\mu(R; W_s)$ is a linear function, i.e., $\hat{s}(R) = W_s R + \Sigma(Y)^{\frac{1}{2}} \boldsymbol{n}$. Then, setting $W_s$ as an instantiate of the orthogonal complement of $W_c$ leads to statistical independence, i.e., $\hat{c}(R) \perp\!\!\!\perp \hat{s}(R)$. Here, $\perp\!\!\!\perp$ denotes the statistical independence, and we define $\langle a, b \rangle_M = \langle a, Mb \rangle$ for a given semi-definite matrix $M$. The orthogonality $A \perp_M B$ of two subspaces $A$ and $B$ is defined likewise.*

*Proof.* Under the assumption in Lemma 1, setting $W_s$ as an instantiate of the orthogonal complement of $W_c$, we have:

$$
\mathrm{ker}(W_s)^{\perp} \perp_M \mathrm{ker}(W_c)^{\perp} \Longleftrightarrow \mathrm{im}(W_s^{\top}) \perp_M \mathrm{im}(W_c^{\top}) \Longleftrightarrow \langle W_s^{\top} \boldsymbol{a}, W_c^{\top} \boldsymbol{b} \rangle_M = 0 \,\forall \boldsymbol{a}, \boldsymbol{b}
$$

$$
\Longleftrightarrow \langle W_s^{\top} \boldsymbol{a}, M W_c^{\top} \boldsymbol{b} \rangle = 0 \,\forall \boldsymbol{a}, \boldsymbol{b} \Longleftrightarrow W_s M W_c^{\top} = 0 \Longleftrightarrow \mathbb{E}_R W_s (R - m)(R - m)^{\top} W_c^{\top} = 0
$$

$$
\Longleftrightarrow \mathbb{E}_{R, \boldsymbol{n}} W_s \left( R + \Sigma^{\frac{1}{2}} \boldsymbol{n} - m \right)(R - m)^{\top} W_c^{\top} = 0 \Longleftrightarrow Cov\left( \hat{s}(R), \hat{c}(R) \right) = 0
$$

$$
\Longleftrightarrow \hat{c}(R) \perp\!\!\!\perp \hat{s}(R)
$$

(17)

$\square$

# D MORE DETAILS ABOUT EVALUATION METRICS AND TRAINING DETAILS

**Evaluation metrics.** For MNIST dataset, we set the maximum perturbation bound $\epsilon = 0.3$, perturbation step size $\eta = 0.01$, and the number of iterations $K = 40$ for PGD and C&W attacks, which keeps the same as (Zhang et al., 2019). Following (Rice et al., 2020), we set perturbation bound $\epsilon = 8/255$, perturbation step size $\eta = \epsilon/10$, and the number of iterations $K = 20$ for CIFAR-10 dataset.

**Training details.** For MNIST, we use the same CNN architecture as (Carlini & Wagner, 2017; Zhang et al., 2019). Following (Zhang et al., 2019), the network is trained using SGD with 0.9 momentum for 50 epochs with an initial learning rate 0.01, and the batch size is set to 128. Hyperparameters used to craft adversarial examples for training are the same as those used for evaluation. These two networks share the same hyper-parameters: we use SGD with 0.9 momentum, weight decay $2 \times 10^{-4}$, batch size 128, and an initial learning rate of 0.1. The maximum epoch is 120, and the learning rate is divided by 10 at epoch 60 and 90, respectively. To generate adversarial examples for training, we set the maximal perturbation $\epsilon = 8/255$, the perturbation step size $\eta = 2/255$, and the number of iterations $K = 10$, which is the same as (Rice et al., 2020). In all of our experiments $\beta$ is set to 1.0. For CausalAdv-M, $\lambda$ is set to 1.0 and 0.5 for CIFAR-10 and CIFAR-100 datasets, respectively. For CausalAdv-T, $\lambda$ is set to 0.5 and 1.0 for CIFAR-10 and CIFAR-100 datasets, respectively.

# E EXPERIMENTS OF WRN-34-10 ON CIFAR-10

Table 6: Classification accuracy (%) of WRN-34-10 on CIFAR-10 under the white-box threat model. The best-performance model and the corresponding accuracy are highlighted.

| Method | Best checkpoint | | | | Last checkpoint | | | |
|---|---|---|---|---|---|---|---|---|
| | Natural | FGSM | PGD-20 | CW-20 | Natural | FGSM | PGD-20 | CW-20 |
| Madry | **86.63** | 59.48 | 53.65 | 53.58 | **86.60** | 57.07 | 49.23 | 49.46 |
| CausalAdv-M | 85.24 | **61.22** | **55.17** | **55.68** | 85.61 | **60.08** | **51.76** | **52.59** |
| TRADES | **84.32** | 60.94 | 56.69 | 54.87 | **84.86** | 59.94 | 52.04 | 52.39 |
| CausalAdv-T | 84.19 | **61.62** | **57.36** | **55.75** | 84.35 | **61.57** | **55.15** | **55.23** |

In Table 6, we report the accuracy of WRN-34-10 (Zagoruyko & Komodakis, 2016) of Madry, TRADES, and the proposed method on CIFAR-10 against various attacks, i.e., FGSM, PGD, and

C&W attacks, which are widely used in the literature. Here, "Natural" denotes the accuracy of natural test images. We denote by PGD-20 the PGD attack with 20 iterations for generating adversarial examples, which also applies to the C&W attack. We can see that the proposed method achieves the best robustness against all three types of attacks, demonstrating that taking into account the spurious correlation can significantly improve the adversarial robustness. Note that the standard deviations of 5 runs are omitted, because they hardly affect the results.

## F ABLATION STUDY

Table 7: Robust accuracy (%) of ResNet-18 on CIFAR-10 and CIFAR-100 under the white-box threat model. For simplicity, we use $t_1$, $t_2$, and $t_3$ to represent the first, second, and third terms in Eq. 11, respectively. The best-performance model and the corresponding accuracy are highlighted.

| Method | $t_1$ | $t_2$ | $t_3$ | CIFAR-10 | | | CIFAR-100 | | |
|---|---|---|---|---|---|---|---|---|---|
| | | | | FGSM | PGD-20 | CW-20 | FGSM | PGD-20 | CW-20 |
| Madry | ✓ | | | 56.69 | 51.92 | 51.00 | 56.69 | 51.92 | 51.00 |
| CausalAdv-M | ✓ | | ✓ | 57.98 | 54.44 | 52.51 | 57.98 | 54.44 | 52.51 |
| TRADES | ✓ | ✓ | | 57.25 | 53.64 | 51.39 | 57.25 | 53.64 | 51.39 |
| CausalAdv-T | ✓ | ✓ | ✓ | **58.97** | **54.55** | **52.95** | **58.97** | **54.55** | **52.95** |

We implicitly conducted ablation studies when designing Table 1, Table 2, and Table 3. To further understand the comparative effects of different terms of the proposed method, we reorganize the robust accuracy of the best checkpoint trained on CIFAR-10 and CIFAR-100 in Table 7. Comparing Madry, TRADES, and CausalAdv-M, we find that introducing the second ($t_2$) and the third term ($t_3$) can improve the robustness and that the effect of these two terms is close. Similarly, comparing TRADES and CausalAdv-T, we see that introducing the third term ($t_3$) can further improve the robustness.

To further analyze the superiority of our method, we compare CausalAdv-M with Madry to explore which kind of adversarial examples that CausalAdv-M is more robust to. These analyses are based on the spurious perspective, as one main conclusion of our paper is that the origin of adversarial vulnerability is the excessive focus of DNNs on spurious correlations between labels and style variables. Specifically, we calculate the KL-divergence between $KL\left(P\left(Y|h\left(x, s\right)\right)||P\left(Y|h\left(x_{adv}, s\right)\right)\right)$ for each input $x$, and divide samples into several (10 in our experiment) bins according to the KL-divergence. Then, we evaluate the robust accuracy of models trained with CausalAdv-M and Madry in each bin sample, and results are depicted in Fig. 3. We can see that CausalAdv-M is more robust

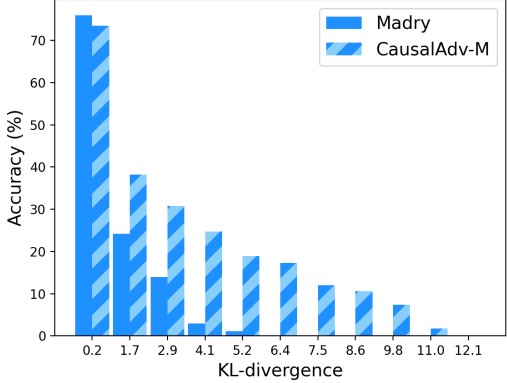

Figure 3: Classification accuracy (%) on CIFAR-10 under the PGD attack with with $\epsilon = 8/255$.

to samples leading to large KL-divergence than Madry. According to these empirical results, we can conclude that the proposed method is more robust to samples causing a significant difference between natural and adversarial distributions.

To further verify that CausalAdv-T can outperform TRADES interms of both the natural and adversairal robustness, we compare the robust accuracy of CausalAdv-T with that of TRADES on CIFAR-10 and CIFAR-100 dataset. The results evaluated on the best checkpoint models are shown in Table 8. We can see that CausalAdv-T can improve both the natural and the adversarial accuracy.

Table 8: Robust accuracy (%) of ResNet-18 on CIFAR-10 and CIFAR-100 under the white-box threat model.

|  | CIFAR-10 | | | | CIFAR-100 | | | |
|---|---|---|---|---|---|---|---|---|
|  | Natural | FGSM | PGD20 | CW20 | Natural | FGSM | PGD20 | CW20 |
| TRADES | 81.39 | 57.25 | 53.64 | 51.39 | 53.85 | 29.04 | 27.91 | 24.09 |
| CausalAdv-T | **81.72** | **58.26** | **54.06** | **51.90** | **53.91** | **30.19** | **28.11** | **24.90** |

To explore the sensitivity of our method on hyperparameters, we evaluate our methods on CIFAR10 and CIFAR100 datasets. The results are organized in Table 9. Results in Table 9 demonstrate that

Table 9: Robust accuracy (%) of ResNet-18 on CIFAR-10 and CIFAR-100 under the white-box threat model.

|  | CIFAR10 | | | | CIFAR100 | | | |
|---|---|---|---|---|---|---|---|---|
|  | Natural | FGSM | PGD20 | CW20 | Natural | FGSM | PGD20 | CW20 |
| Madry | 83.56 | 56.69 | 51.92 | 51.00 | 55.98 | 28.39 | 25.15 | 24.04 |
| CausalAdv-M $\lambda = 0.2$ | 82.17 | 57.13 | 53.28 | 51.80 | 54.68 | 29.85 | 27.32 | 25.88 |
| CausalAdv-M $\lambda = 0.5$ | 80.83 | 57.28 | 53.65 | 51.97 | 54.07 | 29.76 | 27.62 | 25.44 |
| CausalAdv-M $\lambda = 1.0$ | 80.42 | 57.98 | 54.44 | 52.51 | 52.69 | 29.77 | 27.80 | 26.04 |
| CausalAdv-M $\lambda = 1.5$ | 80.17 | 57.53 | 53.24 | 51.40 | 52.37 | 30.74 | 28.49 | 26.71 |
| CausalAdv-M $\lambda = 2.0$ | 80.35 | 58.29 | 53.16 | 52.71 | 48.40 | 30.07 | 28.52 | 25.97 |
| TRADES | 81.39 | 57.25 | 53.64 | 51.39 | 53.85 | 29.04 | 27.91 | 24.09 |
| CausalAdv-T $\lambda = 0.2$ | 81.72 | 58.26 | 54.06 | 51.90 | 53.83 | 29.76 | 28.05 | 24.49 |
| CausalAdv-T $\lambda = 0.5$ | 81.22 | 58.97 | 54.55 | 52.95 | 53.91 | 30.19 | 28.11 | 24.90 |
| CausalAdv-T $\lambda = 1.0$ | 79.65 | 58.48 | 54.45 | 52.89 | 53.17 | 30.66 | 28.57 | 25.74 |
| CausalAdv-T $\lambda = 1.5$ | 78.09 | 57.42 | 53.66 | 51.38 | 52.22 | 30.49 | 28.17 | 25.48 |
| CausalAdv-T $\lambda = 2.0$ | 74.42 | 55.29 | 52.07 | 50.04 | 50.40 | 30.22 | 28.14 | 25.49 |

both CausalAdv-M and CausalAdv-T are relatively insensitive to the hyperparameters.

## G    MORE DETAILS ABOUT ADVERSARIAL LEARNING

Recent work on improving adversarial robustness mainly falls into two categories: certified defense and empirical methods.

Certified defense (Raghunathan et al., 2018; Wong & Kolter, 2018; Singla & Feizi, 2020) aims to endow the model with provably adversarial robustness against norm-bounded perturbations. Although the certified defense strategy is promising, the empirical defense (Goodfellow et al., 2015; Madry et al., 2018; Zhang et al., 2019; Pang et al., 2020; Wong & Kolter, 2018; Xie et al., 2019; Yang et al., 2019), especially the adversarial training method (Goodfellow et al., 2015; Madry et al., 2018; Zhang et al., 2019), is currently the most effective strategy. Empirical defense firstly generates adversarial examples using a certain adversarial attack, then incorporates the generated adversarial examples into the training process. Recently, an empirical detection strategy is to utilize a two-sample test to detect adversarial examples (Gao et al., 2021). In the following, we mainly discuss defense strategies, as the detection approach is not the main focus of this paper.

Recently, various efforts (Najafi et al., 2019; Carmon et al., 2019; Shafahi et al., 2019; Wong et al., 2019; Pang et al., 2020; Rice et al., 2020) have been devoted to improving adversarial training. One line of work focuses on accelerating the training procedure (Shafahi et al., 2019; Wong et al., 2019). Another line of research (Najafi et al., 2019; Carmon et al., 2019) shows a promising direction that unlabeled training data can significantly mitigate the adversarial vulnerability. Lastly, recent work (Pang et al., 2020; Rice et al., 2020) provides an interesting direction where these methods rethink the adversarial training from a exciting aspect, i.e., rethinking the role of normalization (Pang et al.,

2020) and basic training strategies (Rice et al., 2020). However, all these methods overlook the spurious correlation between labels and the style information.

Another related work is (Ilyas et al., 2019), which provides an interesting viewpoint, i.e., adversarial examples can be viewed as a human phenomenon because the model's reliance on useful but not robust features leads to adversarial vulnerability. Our work gives a new causal perspective of adversarial vulnerability. Specifically, a) (Ilyas et al., 2019) found some features were useful but not robust, while our work explores the phenomenon's fundamental cause and provides a clear explanation of why some features are useful but not robust: Given $X$, labels $Y$ are spuriously correlated with the style variables, so fitting the spurious correlation can predict labels. Thus, the style variables can be viewed as 'features'; b) (Ilyas et al., 2019) claimed that adversarial examples could be viewed as a human phenomenon, while our work shows that adversarial examples can be viewed as a model phenomenon rather than merely a human phenomenon. Specifically, the adversarial vulnerability results from fitting the correlation between labels and style variables and failing to fit the causal relations, i.e., DNNs fail to extract content variables.

## H  MORE DETAILS ABOUT CAUSAL REASONING

The most relevant work is CAMA (Zhang et al., 2020a) that aims to improve the robustness of DNNs on unseen perturbation via explicitly modeling the perturbation from a causal view. The main difference between our method and CAMA is that we focus on the adversarial vulnerability while CAMA aims to improve the robustness of unseen perturbations. In addition, CAMA assumes a hard intervention on a latent variable. It promotes robustness via modeling the perturbation in the latent space. In this paper, we employ a soft intervention and propose to penalize DNNs when the adversarial distribution is different from the natural distribution. A recent work (Bühlmann, 2020) also aims to connect robust learning and causality, but the main focus of (Bühlmann, 2020) is on out-of-distribution generalization, which is different from adversarial learning.

Another related work is RELIC (Mitrovic et al., 2020), a regularizer used in self-supervised learning that uses the independence of mechanisms (Peters et al., 2017) and encourages DNNs to be invariant to different augmentations of the same instance. The self-supervised learning method (Mitrovic et al., 2020) also constructs a causal graph to model the data generation process, but the focus of RELIC is on the content invariant property, overlooking the importance of style information. One concurrent work (Tang et al., 2021) propose using the instrumental variable to perform causal intervention, based on a strong assumption that the adversarial vulnerability results from the confounding effect. In contrast, merely some general assumptions are required in this paper, e.g., causal model assumption. Although (Sagawa et al., 2020) takes spurious correlations into account, (Sagawa et al., 2020) proposes using prior knowledge to group the training data to avoid the reliance on spurious correlations. Difference from (Sagawa et al., 2020), our method does not rely on prior knowledge. Another related work is CORE (Heinze-Deml & Meinshausen, 2021), a regularizer inspired by a causal graph is proposed to minimizing the variance of prediction and loss condition on label and ID information to mitigate the influence of domain shift. Different from (Heinze-Deml & Meinshausen, 2021), our method is designed to eliminate the difference between the natural and adversarial distributions.

