# OpenReview forum: "Adversarial Robustness Through the Lens of Causality"
_ICLR.cc/2022/Conference — ICLR 2022 Poster_

### Official Review · Reviewer_Jqu4 · 2021-11-01

**Correctness:** 3
**Technical Novelty And Significance:** 3
**Empirical Novelty And Significance:** 3
**Recommendation:** 8
**Confidence:** 3

**Main Review:**

Strengths
1.	Overall the paper is well written and easy to follow.
2.	The proposed causal graph well models the generation process of adversarial attacks and sheds new light on how to understand and defend against attacks from a causal perspective.
3.	The proposed approach is novel and well-motivated.
4. The authors empirically validate the proposed method using several attacks on prevailing datasets.

Weaknesses
1.	In experiments, the current version only compares with SOTAs using attacks with 20 iterations. It would be helpful if attacks with more iterations are employed as the performance of some defense techniques is shown to drop significantly when the number of attack iterations increases.
2.	Some technical details may need to be clarified. See my questions below.

Questions
 1. In the first term of Eq 4, the authors aim to minimize the divergence between $P(Y|X)$ and $P_{\theta}(Y|\widetilde{X})$. As  $P_{\theta}(Y|\widetilde{X})$ cannot be formulated analytically, the authors introduce $Q_{\theta}$ as an anchor and minimize $KL(P_{\theta}(Y|\widetilde{X})||Q_{\theta}(Y|\widetilde{X}))$ and $KL(P(Y|X)||Q_{\theta}(Y|X))$ instead. However, from my perspective this objective only optimize the anchor instead of pushing $P_{\theta}(Y|\widetilde{X})$ towards $P(Y|X)$, if I am not mistaken.

2. How to connect $g$ and $h$ by sharing the same representation?  It seems that $g$ is a linear function while $h$ is supposed to be parameterized by a neural network.

3. In Eq 8, why $\hat{s}(X)$ can approximate $s(X)$? It seems that there is no supervision signal to ensure this point.

4. If $\hat{s}(X)$ is the integrated representation of both $s$ and $x$, as mentioned in the paper, are $\hat{c}(X)$ and $\hat{s}(X)$ supposed to be independent when $x$ is given?

It would be helpful if these questions can be addressed.


**Summary Of The Paper:**

This paper proposes a causal graph to model the generation process of adversarial attacks. Based on the proposed causal graph, the authors identify the origin of adversarial vulnerability as the spurious correlation between style variable and class label. Under the adversarial distribution, such spurious correlation can be maliciously used to mislead a victim model. In this light, the authors propose a method to align the adversarial distribution and the natural distribution to prevent a model from learning spurious correlation. The proposed method is empirically validated on prevailing datasets under several attacks.

**Summary Of The Review:**

Given the strengths listed, I tend to accept this paper. I suggest the authors concern the questions listed to be addressed.

---

> ### Author Response · Authors · 2021-11-19
> **Response to reviewer Jqu4**
>
> All authors are grateful for your time devoted to reviewing this paper and your constructive suggestions. Here are our detailed replies to your questions.
>
> Q1: “In experiments, the current version only compares with SOTAs using attacks with 20 iterations. It would be helpful if attacks with more iterations are employed as the performance of some defense techniques is shown to drop significantly when the number of attack iterations increases.”
>
> A1:
> Thanks for your valuable suggestion. We evaluate the robustness of the best checkpoint model trained with ADAM on the cifar10 dataset, using PGD and CW attacks with up to 1000 iterations. We can see that the robust accuracy decreases as increasing the iteration, but the robust accuracy of our method under PGD-1000 and CW-1000 attacks is still higher than Madry under the PGD-20 attack, as the robust accuracy of Madry under PGD-20 and CW-20 attacks are 51.92% and 51.00%.
>
> |              |     K=20     |     K=50     |     K=100    |     K=500    |     K=1000    |
> |--------------|--------------|--------------|--------------|--------------|---------------|
> |     PGD-K    |     54.44    |     53.12    |     53.07    |     52.98    |     52.98     |
> |     CW-K     |     52.97    |     51.63    |     51.50    |     51.48    |     51.48     |
>
> Q2: "why optimizing the anchor can push adversarial distributions to natural distribution."
>
> A2:
> Thanks for pointing out the potentially confusing details. We have highlighted the following explanation in our revision.
>
> Here, we give an intuitive explanation of why optimizing the anchor can make the adversarial distribution similar to the natural distribution. Considering that both the adversarial distribution $P_{\theta}\left(Y|\tilde{X}\right)$ and the anchor $Q_{\theta}\left(Y|\tilde{X}\right)$ are parameterized by $\theta$, so optimizing $\theta$ to minimize $KL\left(P_{\theta}\left(Y|\tilde{X}\right)||Q_{\theta}\left(Y|\tilde{X}\right)\right)$ can force the adversarial distribution be similar to the distribution specified by the classifier $h\left(X;\theta\right)$, i.e., $Q_{\theta}\left(Y|\tilde{X}\right)$. That is, optimizing $Q_{\theta}\left(Y|\tilde{X}\right)$ will also change the adversarial distribution $P_{\theta}\left(Y|\tilde{X}\right)$. In addition, minimizing the divergence $KL\left(P\left(Y|X\right)||Q_{\theta}\left(Y|\tilde{X}\right)\right)$ can endow the classifier with the ability to provide a good prediction performance. Thus, minimizing $KL\left(P_{\theta}\left(Y|\tilde{X}\right)||Q_{\theta}\left(Y|\tilde{X}\right)\right)$ will make the adversarial distribution be similar with the natural distribution.
>
> Q3: “How to connect g and h by sharing the same representation? It seems that g is a linear function while h is supposed to be parameterized by a neural network.”
>
> A3:
> Thanks for your valuable question. Your explanation is completely correct, Specifically, $h$ is parameterized by a neural network, where the extracted representation is $r$. And, $g$ is a linear (or non-linear) classifier with $r$ as its input.
>
> Q4: “In Eq 8, why $\hat{s}\left(X\right)$ can approximate $s\left(X\right)$? It seems that there is no supervision signal to ensure this point.”
>
> A4:
> Thanks for your valuable question. To approximate $s(X)$, we employ the conclusion drawn from our causal graph. Specifically, we use the independence between the content variable and the style variable as the supervision to approximate $s(X)$. This is because the content variable is independent of the style variable. Consequently, given the representation space of content variables, we can identify a corresponding sub-space to sample style variables, where all samples in such sup-space are independent of content variables.
>
> Q5: “If $\hat{s}\left(X\right)$ is the integrated representation of both $s$ and $X$, as mentioned in the paper, are $\hat{c}\left(X\right)$
>  and $\hat{s}\left(X\right)$ supposed to be independent when $X$ is given?”
>
> A5:
> Thanks for pointing out this potentially confusing point. We have added the following explanation to our revision.
>
> The underlying intuition is that we aim to model the causal relation between the content variable and the style variable and omit the spurious correlation between $\hat{c}\left(X\right)$ and $\hat{s}\left(X\right)$, because the spurious correlation is not a causal relation, although $\hat{c}\left(X\right)$ and $\hat{s}\left(X\right)$ are not statistically independent.
>
> Following your constructive suggestion, we have revised our paper. Some changes are:
>
> Appendix A, the second paragraph: ``Here, we give an intuitive explanation of why … ‘’.
>
> Sec. 3.3, the last paragraph: ``The underlying intuition is that … ’’.

---

> > ### Comment · Reviewer_Jqu4 · 2021-11-29
> > **Thank you for your response**
> >
> > Thank the authors for the response. Most of my concerns have been addressed.

---

### Official Review · Reviewer_iRoe · 2021-11-02

**Correctness:** 4
**Technical Novelty And Significance:** 4
**Empirical Novelty And Significance:** 4
**Recommendation:** 8
**Confidence:** 4

**Main Review:**

Strengths:
* This paper provides a new perspective on analyzing adversarial robustness and it is novel for me.
* This paper is well organized and easy to follow.
* The distribution alignment method inspired by theory shows good robustness

Weakness:
* I noticed the Natural accuracy on CIFAR-10 and CIFAR-100 can sometimes be worse than TRADES and Madry. Given the trade-off between natural and adversarial accuracy is adjustable, it would be good to adjust the trade-off to see if the proposed method can simultaneously surpass TRADES in terms of both natural accuracy and robustness.
* How to select the hyper-parameters $\lambda$, $\beta$ and $\gamma$. Are they sensitive? Are they consistent among different datasets?

Detail comments:
* First paragraph of Page 9, "we report robust accuracy of WRN-34-10 trained with CIFAR10 dataset on Auto-Attack, Madry: 49.58%, ADA-M: 51.56%, TRATDES: 52.46%" TRATDES --> TRADES
* Some "Mardry" should be "Madry"

**Summary Of The Paper:**

This paper presents a causal perspective on addressing adversarially vulnerability. It first constructs a causal graph, which then inspires the design of the distribution alignment method for reducing the gap between adversarial and natural data. Extensive experiments on CIFAR10, CIFAR100, and MNIST demonstrate the robustness of the proposed method against various attack methods.

**Summary Of The Review:**

I enjoy reading the analysis of the paper. This new perspective is novel to me and I would tend to accept this paper. If the author can address my concerns on the experiments. I would be more convinced.

---

> ### Author Response · Authors · 2021-11-19
> **Response to reviewer iRoe [1/3]**
>
> All authors are grateful for your time devoted to reviewing this paper and your constructive suggestions. Here are our detailed replies to your questions.
>
> Q1: “Given the trade-off between natural and adversarial accuracy is adjustable, it would be good to adjust the trade-off to see if the proposed method can simultaneously surpass TRADES in terms of both natural accuracy and robustness.”
>
> A1:
> Thanks for your constructive suggestion! Following your suggestion, we compare ADAT with TRADES on CIAFR10 and CIFAR100, where the robustness is evaluated on the best checkpoint model. The results are as follows, and the corresponding discussion has been added to our revision.
>
> Comparison between ADA-T and TRADES on the CIFAR-10 dataset.
>
> |     CIFAR10    |     Natural    |     FGSM     |     PGD20    |     CW20     |
> |----------------|----------------|--------------|--------------|--------------|
> |     TRADES     |     81.39      |     57.25    |     53.64    |     51.39    |
> |     ADA-T      |     81.72      |     58.26    |     54.06    |     51.90    |
>
> Comparison between ADA-T and TRADES on the CIFAR-100 dataset.
>
> |     CIFAR100    |     Natural    |     FGSM     |     PGD20    |     CW20     |
> |-----------------|----------------|--------------|--------------|--------------|
> |     TRADES      |     53.85      |     29.04    |     27.91    |     24.09    |
> |     ADA-T       |     53.91      |     30.19    |     28.11    |     24.90    |
>
> We can see that, compare to TRADES, the proposed ADAT can improve both the natural and adversarial accuracy, which is consistent with your intuition. These results further demonstrate the necessity of considering the spurious correlation when developing robust models.
> All the authors believe that the solid suggestion would further improve our paper.

---

> > ### Comment · Reviewer_iRoe · 2021-11-22
> > **My concerns have been addressed**
> >
> > The authors' excellent rebuttal addresses all my concerns and I have improved my score to 8.

---

> > > ### Author Response · Authors · 2021-11-23
> > > **Thank you very much**
> > >
> > > Dear Reviewer iRoe,
> > >
> > > Thank you very much for increasing the score. We are delighted to see that our response has addressed your concerns. Thanks again for your acceptance recommendation.
> > >
> > > Best Regards,
> > >
> > > Authors

---

> ### Author Response · Authors · 2021-11-19
> **Response to reviewer iRoe [2/3]**
>
> Q2: “How to select the hyper-parameters \lambda, \beta and \gamma. Are they sensitive? Are they consistent among different datasets?”
>
> A2:
> Thanks for your valuable questions. We have added the following settings into our revision.
>  - Consideration of consistency:
> In all of our experiments, $\beta$ is set to 1.0. For ADA-M, we set $\gamma=0$, and $\lambda$ is selected as follows:
> In our experiments for CIFAR-10, $\lambda$ is set to 1.0, and $\lambda$ is set to 0.5 for CIFAR-100. For ADA-T, similar to TRADES, we set $\gamma=1/6$, and $\lambda$ is selected as follows: In our experiments for CIFAR-10, $\lambda$ is set to 0.5 and $\lambda$ is set to 1.0 for CIFAR-100.
>
>  - Consideration of sensitivity:
> In addition, following your suggestion, we evaluate the sensitivity of our method on these parameters on CIFAR-10 and CIFAR-100 datasets. The following results have been added to our revision.
>
> Evaluating the sensitivity of ADA-M on the CIFAR-10 dataset.
>
> |     CIFAR10             |     Natural    |     FGSM     |     PGD20    |     CW20     |
> |-------------------------|----------------|--------------|--------------|--------------|
> |     Madry               |     83.56      |     56.69    |     51.92    |     51.00    |
> |     ADA-M $\lambda=0.2$    |     82.17      |     57.13    |     53.28    |     51.80    |
> |     ADA-M $\lambda=0.5$    |     80.83      |     57.28    |     53.65    |     51.97    |
> |     ADA-M $\lambda=1.0$    |     80.42      |     57.98    |     54.44    |     52.51    |
> |     ADA-M $\lambda=1.5$    |     80.17      |     57.53    |     53.24    |     51.40    |
> |     ADA-M $\lambda=2.0$    |     80.35      |     58.29    |     53.16    |     52.71    |
>
> Evaluating the sensitivity of ADA-T on the CIFAR-10 dataset.
>
> |     CIFAR10              |     Natural    |     FGSM     |     PGD20    |     CW20     |
> |--------------------------|----------------|--------------|--------------|--------------|
> |     TRADES               |     81.39      |     57.25    |     53.64    |     51.39    |
> |     ADA-T $\lambda=0.2$    |     81.72      |     58.26    |     54.06    |     51.90    |
> |     ADA-T $\lambda=0.5$       |     81.22      |     58.97    |     54.55    |     52.95    |
> |     ADA-T $\lambda=1.0 $      |     79.65      |     58.48    |     54.45    |     52.89    |
> |     ADA-T $\lambda=1.5 $      |     78.09      |     57.42    |     53.66    |     51.38    |
> |     ADA-T $\lambda=2.0 $     |     74.42      |     55.29    |     52.07    |     50.04    |
>
> Evaluating the sensitivity of ADA-M on the CIFAR-100 dataset.
>
> |     CIFAR100            |     Natural    |     FGSM     |     PGD20    |     CW20     |
> |-------------------------|----------------|--------------|--------------|--------------|
> |     Madry               |     55.98      |     28.39    |     25.15    |     24.04    |
> |     ADA-M $\lambda=0.2$    |     54.68      |     29.85    |     27.32    |     25.88    |
> |     ADA-M $\lambda=0.5$    |     54.07      |     29.76    |     27.62    |     25.44    |
> |     ADA-M $\lambda=1.0$    |     52.69      |     29.77    |     27.80    |     26.04    |
> |     ADA-M $\lambda=1.5$    |     52.37      |     30.74    |     28.49    |     26.71    |
> |     ADA-M $\lambda=2.0$    |     48.40      |     30.07    |     28.52    |     25.97    |
>
> Evaluating the sensitivity of ADA-T on the CIFAR-100 dataset.
>
> |     CIFAR100          |     Natural    |     FGSM     |     PGD20    |     CW20     |
> |-----------------------|----------------|--------------|--------------|--------------|
> |     TRADES            |     53.85      |     29.04    |     27.91    |     24.09    |
> |     ADA-T $\lambda=0.2$    |     53.83      |     29.76    |     28.05    |     24.49    |
> |     ADA-T $\lambda=0.5 $   |     53.91      |     30.19    |     28.11    |     24.90    |
> |     ADA-T $\lambda=1.0$    |     53.17      |     30.66    |     28.57    |     25.74    |
> |     ADA-T $\lambda=1.5 $   |     52.22      |     30.49    |     28.17    |     25.48    |
> |     ADA-T $\lambda=2.0 $   |     50.40      |     30.22    |     28.14    |     25.49    |
>
> Thanks for your constructive suggestion, all authors believe that your valuable comments would further improve our paper.

---

> ### Author Response · Authors · 2021-11-19
> **Response to reviewer iRoe [3/3]**
>
> Q3: “Detail comments: First paragraph of Page 9: TRATDES --> TRADES. Some "Mardry" should be "Madry"”
>
> A3: Thank you for your patience and careful attention to pointing out these typos. We have fixed all of these typos in our revision.
>
> Following your constructive suggestion, we have revised our paper. Some changes are:
>
> Appendix D, Training details.
>
> Appendix F, Figure 8, and Table 9 and their corresponding explanation.

---

### Official Review · Reviewer_aWQa · 2021-11-03

**Correctness:** 3
**Technical Novelty And Significance:** 3
**Empirical Novelty And Significance:** 2
**Recommendation:** 6
**Confidence:** 4

**Details Of Ethics Concerns:**

I do not foresee any ethical concerns in the proposed methods. Adversarial robust learning, as studied, enables safer deployment of machine learning models.

**Main Review:**

The work provides a plausible causal representation of the adversarial attack process and derives a useful explanation for vulnerability. The robust approach derived from their explanation is quite reasonable and is novel to my knowledge. It shows improvement on MNIST, CIFAR-10, and CIFAR-100 datasets. However, the confidence intervals are omitted which makes the results less convincing.
The choice of comparing against only two baselines, Madry and TRADES, can be better motivated, especially since many alternative methods exists. Consider comparing against the top performing method from RobustBench (https://robustbench.github.io/).
Scope of the analysis is not well defined. The types of adversarial attacks, types of classifiers, and types of data are not specified, and implicitly assumed to be image-based neural networks.
The description of the causal graph makes it hard to understand that two processes are being represented - the natural data distribution and its relation to the distribution of images constructed from adversarial attack for a given classifier(s). The robust learning method based on adversarial alignment is not explained well and involves choices that are unjustified. Statements on relationship of the proposed method to Madry et al. 2017 and Zhang et al. 2019 are imprecise.


# Related work

Admittedly the literature on this problem area is vast, however, the following two related works will be important to discuss.
A causal graph motivated adversarial learning approach is proposed in Heinze-Deml and Meinshausen 2021 (Conditional variance penalties and domain shift robustness. Machine Learning https://link.springer.com/article/10.1007/s10994-020-05924-1). The graph has core (content) and style nodes although it is different from the one proposed in this work.
The relationship between causal and adversarial approaches to robust learning was discussed in Bühlmann 2020 (Invariance, Causality and Robustness. Statistical Science https://arxiv.org/abs/1812.08233). The discussion is for a different causal graph where X causes Y and different type of adversarial perturbations that are unbounded, however, the work is related as it also gives a causal interpretation to robust learning.
Although non-causal in motivation, other robust learning work has also identified spurious correlation between labels and style features as reason for lack of generalizability. For example, see Sagawa et al. 2020 (Distributionally Robust Neural Networks for Group Shifts. ICLR https://arxiv.org/abs/1911.08731). Consider discussing this line of work into out-of-distribution generalization.


# Questions that I would like the authors to respond to:

What is the scope of domains for the proposed causal graph in Figure 1? Is it applicable to image or any high-dimensional data problems? Please specify the problems for which this causal graph is suitable. The graph makes unstated assumptions like Y is an effect not a cause of C (i.e. anti-causal learning) which should be motivated in context of some problem domains.

Why should the adversarial distribution be a result of an intervention on some causal graph, where the graph is shared with the one generating the natural distribution? This assumption does not seem necessary as the defintion of adversarial perturbation E_adv in Eq (2) does not rely on the causal graph. Please clarify if this is not a limiting assumption.

What are type of adversarial attacks are in scope? This should be specified beforehand. The argument about origin of adversarial vulnerability relies on the assumption that conditional distribution of style P(s | X) does not vary much with adversarial attacks. However, some attacks change the image drastically such as ones that add image patches.


# Questions below are minor and do not seriously affect my review:

Can you test your hypothesis that spurious correlation between labels and style features is the reason for adversarial vulnesrabitliy such as by studying known attacks? This will provide more evidence to the analysis of the assumed causal graph.

The relation between the two causal models \mathcal{M} and \mathcal{M}_adv is not clear. Mention the specific hard or soft intervention that results in the latter one like do(E\sim E_adv).

On the specific way of constructing adversarial distribution in Eq (2), the X and E are added instead of any other general function. Please state if this is a necessary assumption.

Can you comment on the order of the standard deviations for numbers in Table 1, 2, 3? Ideally, these should be included in Appendix even if small.


# Suggestions for improving the writing:

The statement claiming that objective in Eq. (7) is same as in Madry et al. 2017 for \lambda and \gamma equal to 0 is incorrect. The objective in Madry et al. solves a min-max problem maximising loss over perturbations of natural data points. Similarly, TRADES Zhang et al. 2019 also has a term containing maximum loss over perturbations.

The term nuisance factors is used in Sections 1,2 without defining it until later using the style features. Consider briefly mentioning an example of nuisance factors early on.

The term integrated representation s(X) in Eq (6) is not clear. Please give an example of such a function.

I did not understand how Footnote 3 is an explanation for ignoring dependence between C and S. Even if the relation is not causal, it can be modelled as correlation exists based on the causal graph.

Relationship with the work of Ilyas et al. 2019 on the origin of adversarial vulnerability can be moved to the main text from Appendix F as it is important to highlight what additional insights does a causal perspective provides.

Consider mentioning the omitted explanation for change in P(Y | \tilde{X}, s) since the path from S to Y given \tilde{X} is open due to conditioning on a child of the collider X.

Sum notation in Eq (3) should clarify that \mathbb{S} is discrete.

The remark from Gopnik et al. 2004 in the Introduction can be made more precise to the specific experimental conditions of that work instead of a claim for all of human cognition.

**Summary Of The Paper:**

The work presents a causal perspective of adversarial attacks on image-based machine learning models by studying a causal graph of the adversarial data creation process and highlighting how such a process makes the learned models vulnerable. It argues that the main reason for adversarial vulnerability is the reliance of models on spurious correlations between labels and style. Accordingly, it proposes a method to learn models for which the conditional distribution of label given style and image does not vary much when attacked. Empirically, the method is shown to be more robust than two baselines on three datasets.

**Summary Of The Review:**

The paper undertakes an original approach to studying the important problem of adversarial vulnerability. However, the description of the method, its design choices, and evaluation requires improvement. That said, the results are quite encouraging and authors should provide more evidence for their hypothesis on reasons for adversarial vulnerability and test their method against stronger baselines.

---
After the response

Most of my concerns have been adequately addressed. The description of the method needs further clarification. I would encourage authors to clarify the choices made in Appendix A. Given that the approach to adversarial robustness is novel, empirically useful, and potentially will inspire further work, I have substantially improved the score and leaning towards a recommendation to Accept.

---

> ### Author Response · Authors · 2021-11-19
> **Response to reviewer aWQa [1/3]**
>
> We appreciate your time and suggestions! Here are our detailed replies to your questions.
>
> 1. Response to “Related work”
>
> Q1: “Admittedly the literature on this problem area is vast, however, the following two related works [1, 2] will be important to discuss.”
>
> A1:
> We thank the reviewer for pointing out these valuable works. As pointed by the reviewer that the literature on this problem area is vast, merely discussing the most related papers may inadvertently miss some valuable work. We have added the following discussion into our revision.
>
> “One related work is CORE [1], a regularizer inspired by a causal graph is proposed to make models robust against domain shift. Specifically, [1] proposes minimizing the variance of prediction and loss conditions on label and ID information. Different from [1], our method is designed to eliminate the difference between the natural and adversarial distributions. A recent work [2] aims to connect robust learning and causality, but the main focus of [2] is on out-of-distribution generalization, which is different from adversarial learning. Another related work is [3], where spurious correlations are considered. Specifically, [3] proposes using prior knowledge to group the training data to avoid the reliance on spurious correlations. Difference from [3], our method does not rely on prior knowledge.”
>
> The problem of out-of-distribution (OOD) generalization is different from the adversarial robustness considered in our paper. This is because the test phase environment cannot be observed under the OOD setting, but in the literature of adversarial robustness, adversarial examples can be observed in the training phase. Therefore, we omit the discussion of this line of work.
>
> 2. Response to “Questions that I would like the authors to respond to”
>
> Q2.1: “What is the scope of domains for the proposed causal graph in Figure 1? Is it applicable to image or any high-dimensional data problems? Please specify the problems for which this causal graph is suitable. The graph makes unstated assumptions like Y is an effect not a cause of C (i.e. anti-causal learning) which should be motivated in context of some problem domains.”
>
> A2.1:
>  - The causal graph is constructed for analyzing adversarial vulnerability.
>  - We assume that the perceived data is generated from content and style variables when constructing the causal graph. Although no further assumptions are considered, only image datasets are used for evaluating the proposed method, so the graph is applicable to image data.
>  - Considering the widely observed phenomenon that unlabeled data is helpful for learning, we can assume that the relationship between images and labels is anti-causal. We refer the reviewer to [4,5,6,7] for detailed theoretical and empirical supports.
>
> We have highlighted the scope of the graph in our revision: The causal graph is constructed for modeling the image data generation process and analyzing adversarial vulnerability.
>
>  Q2.2: “Why should the adversarial distribution be a result of an intervention on some causal graph, where the graph is shared with the one generating the natural distribution? This assumption does not seem necessary as the definition of adversarial perturbation E_adv in Eq (2) does not rely on the causal graph. Please clarify if this is not a limiting assumption.”
>
> A2.2:
>  - Our paper aims to connect causality with adversarial vulnerability, and we give a perspective, i.e., considering the difference between adversarial and natural distributions. Thus, we stated in the paper that: “In the context of adversarial learning, we desire the causal graph by which both the natural and the adversarial distributions can be generated”. Consequently, we construct a causal graph, which can generate both the adversarial and the natural distribution.
>  - We model the generation process of $E_{adv}$ by writing $E_{adv}$ as a mechanism $\mathcal{M}_{adv}$ of $X$, $Y$, and $\theta$, i.e., Eq. (2), which is consistent with the constructed causal graph.
>
> Q2.3: “What are type of adversarial attacks are in scope? This should be specified beforehand. The argument about origin of adversarial vulnerability relies on the assumption that conditional distribution of style P(s| X) does not vary much with adversarial attacks. However, some attacks change the image drastically such as ones that add image patches.”
>
> A2.3:
>  - Thanks for pointing out the potentially confusing unstated assumption. We have highlighted the considered adversarial attacks as follows.
> “In this paper, we merely consider the imperceptible adversarial attacks, i.e, the widely studied \ell_{p} norm bounded adversarial examples.”
>  - As mentioned by the reviewer, it is interesting to explore the connection between causality and more general adversarial attacks, e.g., adversarial patch [8] and pixel attack [9]. We leave it as our future work.

---

> ### Author Response · Authors · 2021-11-19
> **Response to reviewer aWQa [2/3]**
>
> 3. Response to “Questions below are minor and do not seriously affect my review”
>
> Q3.1: “Can you test your hypothesis that spurious correlation between labels and style features is the reason for adversarial vulnesrabitliy such as by studying known attacks? This will provide more evidence to the analysis of the assumed causal graph.”
>
> A3.1:
> The spurious correlation is the key to understanding and mitigating adversarial venerability, which is the conclusion drawn from the constructed causal graph, see Sec. 3.1. In addition, our experiments can verify the effectiveness of considering the spurious correlation when designing a robust model, see Sec 4.
>
> Q3.2: “The relation between the two causal models \mathcal{M} and \mathcal{M}_adv is not clear. Mention the specific hard or soft intervention that results in the latter one like do(E\sim E_adv).”
>
> A3.2:
> $\mathcal{M}$ is the mechanism of generating perturbation $E$. $\mathcal{M}_{adv}$ is used for modifying $\mathcal{M}$ to make the generated perturbation be adversarial.
>
> Q3.3: “On the specific way of constructing adversarial distribution in Eq (2), the X and E are added instead of any other general function. Please state if this is a necessary assumption.”
>
> A3.3:
> Following the previous suggestions of the reviewer, i.e., Q2.1, Q2.2, and Q2.3, we are prone to specify the function in our revision. That is, we employ a specific approach to construct adversarial distribution, but we also highlight that the mechanism $\mathcal{M}_{adv}$ can be instantiated by any functions that can generate adversarial examples.
>
> Q3.4: “Can you comment on the order of the standard deviations for numbers in Table 1, 2, 3? Ideally, these should be included in Appendix even if small.”
>
> A3.4:
> Thanks for your valuable suggestion. Following your suggestion, we have added these results to our revision.

---

> ### Author Response · Authors · 2021-11-19
> **Response to reviewer aWQa [3/3]**
>
> 4. Response to “Suggestions for improving the writing”
>
> Q4.1: “The statement claiming that objective in Eq. (7) is same as in Madry et al. 2017 for \lambda and \gamma equal to 0 is incorrect. The objective in Madry et al. solves a min-max problem maximising loss over perturbations of natural data points. Similarly, TRADES Zhang et al. 2019 also has a term containing maximum loss over perturbations.”
>
> A4.1:
> $X+E_{adv}$ stands for adversarial example, which is obtained by maximizing loss over perturbation of natural data point, see Eq (2) and Sec. 3.2. Thus, our statement is correct.
>
> Q4.2: “The term nuisance factors is used in Sections 1,2 without defining it until later using the style features. Consider briefly mentioning an example of nuisance factors early on.”
>
> Q4.2:
> We give the explanation of nuisance factors when we first used this concept, see Sec. 1.
>
> Q4.3: “The term integrated representation s(X) in Eq (6) is not clear. Please give an example of such a function.”
>
> A4.3:
> We use a neural network to instantiate it in this paper, see Sec. 3.3.
>
> Q4.4: “I did not understand how Footnote 3 is an explanation for ignoring dependence between C and S. Even if the relation is not causal, it can be modelled as correlation exists based on the causal graph.”
>
> A4.4:
> We aim to model the causal relationship between the content variable and the style variable and omit the spurious correlation between $\hat{c}\left(X\right)$ and $\hat{s}\left(X\right)$, although $\hat{c}\left(X\right)$ and $\hat{s}\left(X\right)$ are not statistically independent.
>
> Q4.5: “Relationship with the work of Ilyas et al. 2019 on the origin of adversarial vulnerability can be moved to the main text from Appendix F as it is important to highlight what additional insights does a causal perspective provides.”
>
> A4.5: Thanks for your suggestion. We have revised the paper according to your suggestion.
>
> Q4.6: “Consider mentioning the omitted explanation for change in P(Y | \tilde{X}, s) since the path from S to Y given \tilde{X} is open due to conditioning on a child of the collider X.”
>
> A4.6: Thanks for your suggestion.
>
> Q4.7: “Sum notation in Eq (3) should clarify that \mathbb{S} is discrete.”
>
> A4.7: Thanks for your valuable suggestion. Following your valuable suggestion, we have added it into our revision.
>
> Q4.8: “The remark from Gopnik et al. 2004 in the Introduction can be made more precise to the specific experimental conditions of that work instead of a claim for all of human cognition.”
>
> A4.8: Thanks for your suggestion.
>
> [1] Conditional variance penalties and domain shift robustness. Machine Learning, 2021.
>
> [2] Invariance, Causality and Robustness. Statistical Science, 2020.
>
> [3] Distributionally robust neural networks for group shifts: On the importance of regularization for worst-case generalization. ICLR, 2020.
>
> [4] On causal and anticausal learning. ICML, 2012.
>
> [5] Inference of cause and effect with unsupervised inverse regression. AISTATS, 2015.
>
> [6] A Simple Framework for Contrastive Learning of Visual Representations. ICML, 2020.
>
> [7] Unlabeled data improves adversarial robustness. NeurIPS, 2019.
>
> [8] Adversarial patch. NIPS, 2017.
>
> [9] One Pixel Attack for Fooling Deep Neural Networks. IEEE Transactions on Evolutionary Computation.
>
> We have revised our paper following your suggestions. Some changes are:
>
> Appendix H, the second paragraph: ``A recent work … ’’.
>
> Sec. 2, the third paragraph: `` Specifically, we construct a causal graph … ’’.
>
> Sec. 3.1, the second paragraph: `` where we assume … ’’.
>
> Sec. 3.1, the last paragraph: `` Hence, the origin of adversarial vulnerability … ’’.

---

> ### Author Response · Authors · 2021-11-22
> **Would you mind checking our response? Thanks!**
>
> Dear Reviewer aWQa,
>
> Thanks for the comments! We have tried our best to address all the concerns and provided explanations to all questions. Here is a summary of our detailed response below. We humbly expect you can check it and confirm whether our response has addressed your concerns.
>
> - We have highlighted the scope of the proposed causal graph in our revision, i.e., we construct the causal graph for modeling the image data generation process and analyzing adversarial vulnerability.
>
> - We have explained why we regard the adversarial distribution as an intervention distribution.
>
> - We have highlighted which type of adversarial attacks is considered in our work.
>
> - We have revised the paper following your suggestions in both ‘Questions below are minor and do not seriously affect my review’ and ‘Suggestions for improving the writing.’
>
> Thanks a lot for your feedback! Any further discussion/question is welcomed! Your support for a novel, simple, and initial attempt to think of adversarial vulnerability from a causal perspective is very important and we sincerely appreciate it!
>
> Best Regards,
>
> Authors

---

> ### Author Response · Authors · 2021-11-23
> **Would you mind confirming if you have further concerns? Thanks!**
>
> Dear Reviewer aWQa,
>
> We appreciate your comments and time! We have addressed all your questions with a summary and a detailed response below and revised the paper following your suggestion. Would you mind checking it and confirming if you have further questions?
>
> Best Regards,
>
> Authors

---

> ### Author Response · Authors · 2021-11-24
> **[Last six days reminder] Would you mind confirming if you have further questions? Thanks!**
>
> Dear Reviewer aWQa,
>
> We appreciate your comments and time! We have revised the paper following your suggestions. Would you mind checking it and confirming if you have further questions?
>
> Best Regards,
>
> Authors

---

> ### Author Response · Authors · 2021-11-25
> **[Last five days reminder] Would you mind confirming if you have further questions? Thanks!**
>
> Dear Reviewer aWQa,
>
> We appreciate your comments and time! We have revised the paper following your suggestions. Would you mind checking it and confirming if you have further questions?
>
> Best Regards,
>
> Authors

---

> ### Author Response · Authors · 2021-11-26
> **[Last four days reminder] Would you mind confirming if you have further questions? Thanks!**
>
> Dear Reviewer aWQa,
>
> We appreciate your comments and time! We have revised the paper following your suggestions. Would you mind checking it and confirming if you have further questions?
>
> Best Regards,
>
> Authors

---

> ### Author Response · Authors · 2021-11-27
> **[Last three days reminder] Would you mind confirming if you have further questions? Thanks!**
>
> Dear Reviewer aWQa,
>
> We appreciate your comments and time! We have revised the paper following your suggestions. Would you mind checking it and confirming if you have further questions?
>
> Best Regards,
>
> Authors

---

### Official Review · Reviewer_65Tx · 2021-11-05

**Correctness:** 4
**Technical Novelty And Significance:** 4
**Empirical Novelty And Significance:** 3
**Recommendation:** 8
**Confidence:** 3

**Main Review:**

The paper has a lot of strengths:
* While the relationship between causality and adversarial robustness is often stated, this is the first paper I've seen that formally characterizes it. I like the simple separation of content and style variables that allows this characterization.
* The paper derives a method based on the graph criterion. I am not sure about all the assumptions used here (e.g., independent gaussians), but at least on the benchmarks reported, the method performs well.
* The authors provide an interpretation of past methods within their framework, which I thought was a nice unifying insight.

Overall, the strength of this paper is in the formulation of adversarial examples as a causal problem. This provides a different view and motivates better methods for modeling the style-based spurious correlations. My feedback for the authors is:
* Section 3.1: While the theoretical justification makes sense, can you provide a (toy) empirical example to show how the learnt spurious correlation leads to adversarial examples, and not P(x|s)? I get the intuition, but the claim is vague. If you can provide one toy example supporting the claim, it will be stronger.
* Section 3.2:
 --I do not see how Eqn 5 is derived. Are you using something similar to the triangle inequality for the divergences? In that case, will it be an upper bound. Also I do not think KL divergence supports triangle inequality. So this approximation step seems arbitrary--can you justify it?
 --Does it mean that g and h are the same models? Is there any justification for this choice, besides convenience?
* Section 4.2: Can you provide examples of the kind of adversarial examples that ADA method is robust to, which prior work is not? That will be provide more intuition on what exactly is ADA method capturing. Right now, we see that the accuracy increases but not sure why.



**Summary Of The Paper:**

The paper shows a causal perspective to the adversarial robustness problem. It creates a graph over content and style variable sets. It identifies the spurious correlation between style and label as the main reason for adversarial examples, and then proposes a method to remove it from the trained model. Experiments on three datasets show that the proposed method is better than two baselines.

**Summary Of The Review:**

Great paper connecting causality and adversarial robustness

---

> ### Author Response · Authors · 2021-11-19
> **Response to reviewer 65Tx [1/3]**
>
> All authors are grateful for your time devoted to reviewing this paper and your constructive suggestions. Here are our detailed replies to your questions.
>
> Q1: “While the theoretical justification makes sense, can you provide a (toy) empirical example to show how the learnt spurious correlation leads to adversarial examples, and not P(x|s)? I get the intuition, but the claim is vague. If you can provide one toy example supporting the claim, it will be stronger.”
>
> A1:
> Thanks for your valuable suggestion. In the context of imperceptible adversarial examples considered in this paper, neither P(x|s) nor P(s|x) leads to adversarial examples. The reasons are as follows, which has been added to our revision.
> In this work, we focus on imperceptible adversarial examples, where drastic style changes hardly appear. Consequently, adversaries exploit the conditional association between Y and S. Considering that changing the style of samples, i.e., (P(s|x)), can make adversarial perturbations perceptible. Hence, adversaries need to identify perturbations where the conditional association between labels and styles is different from that of natural distributions.

---

> ### Author Response · Authors · 2021-11-19
> **Response to reviewer 65Tx [2/3]**
>
> Q2: “I do not see how Eqn 5 is derived. Are you using something similar to the triangle inequality for the divergences? In that case, will it be an upper bound. Also I do not think KL divergence supports triangle inequality. So this approximation step seems arbitrary--can you justify it? --Does it mean that g and h are the same models? Is there any justification for this choice, besides convenience?”
>
> A2:
> Thanks for pointing out the potentially confusing derivation of Eq. (5). We have added the following explanation to our revision.
> Without loss of generality, the metric d in Eq. (4) can be realized as total variation distance (TVD). Thus, according to the definition of TVD, we have
> $d\left(P\left(Y|X\right), P_{\theta}\left(Y|\tilde{X}\right)\right) \leq d\left(P_{\theta}\left(Y|\tilde{X}\right), Q_{\theta}\left(Y|\tilde{X}\right)\right) + d\left(P\left(Y|X\right), Q_{\theta}\left(Y|\tilde{X}\right)\right)$
>
> where $Q_{\theta}(Y|X)$ is the conditional distribution specified by the classifier $h\left(X;\theta\right)$.
> According to the Pinsker inequality, i.e., $d\left(P,Q\right) \leq \sqrt{\frac{KL\left(P||Q\right)}{2}}$, where $KL$ is the Kullback-Leibler divergence, we have the following upper bound:
>
> $d\left(P\left(Y|X\right), P_{\theta}\left(Y|\tilde{X}\right)\right) \leq
> 	\sqrt{\frac{KL\left(P_{\theta}\left(Y|\tilde{X}\right)|| Q_{\theta}\left(Y|\tilde{X}\right)\right)}{2}} +
> 	\sqrt{\frac{KL\left(P\left(Y|X\right)|| Q_{\theta}\left(Y|\tilde{X}\right)\right)}{2}}$
>
> Now, we can employ the upper bound as a surrogate loss. In practice, we can replace $\sqrt{\frac{KL\left(P||Q\right)}{2}}$ with $KL\left(P||Q\right)$. The intuition is that the difference between $\sqrt{\frac{KL\left(P||Q\right)}{2}}$ and $KL\left(P||Q\right)$ is relatively small when $KL\left(P||Q\right)$ is not extremely large, so the replacement will not introduce much difference. Thus, we have:
> $
> 	KL\left(P_{\theta}\left(Y|\tilde{X}\right)|| Q_{\theta}\left(Y|\tilde{X}\right)\right) \nonumber +
> 	\gamma KL\left(P\left(Y|X\right) || Q_{\theta}\left(Y|X\right)\right),
> $
>
> where $\gamma$ is a tunable hyperparameter. To verify the replacement will not introduce much difference, we evaluate the robustness of ADA-M and ADA-T on the CIFAR-10 dataset with two losses, i.e., $KL\left(P||Q\right)$ and $\sqrt{\frac{KL\left(P||Q\right)}{2}}$. The results evaluated on best checkpoints are shown in the following table. Here, * means that $\sqrt{\frac{KL\left(P||Q\right)}{2}}$ rather than  $KL\left(P||Q\right)$ is used for optimization.
>
> |               |     Natural    |     FGSM     |     PGD20    |     CW20     |
> |---------------|----------------|--------------|--------------|--------------|
> |     Madry     |     83.56      |     56.69    |     51.92    |     51.00    |
> |     ADA-M     |     80.42      |     57.98    |     54.44    |     52.51    |
> |     ADA-M*    |     80.03      |     57.47    |     52.98    |     52.72    |
> |     TRADES    |     81.39      |     57.25    |     53.64    |     51.39    |
> |     ADA-T     |     81.22      |     58.97    |     54.55    |     52.95    |
> |     ADA-T*    |     79.94      |     58.46    |     54.12    |     52.28    |
>
> Model g is a neural network using $s$ to predict labels, and model $h$ is a neural network used for extracting features. Thus, $g$ and $h$ are different models. To ensure that model $g$ can contribute to the representation learning of model $h$, we apply model $g$ to the representation of model $h$. Simply setting $g$ to a linear model works well in our experiments. Following your suggestion, we compare different realizations of mode $g$, i.e., linear mapping and non-linear neural networks. The following results suggest that our method is robust to the selection of model $g$. We have added these results to our revision. Thanks for your valuable suggestion!
>
> |                           |     Natural    |     FGSM     |     PGD20    |     CW20     |
> |---------------------------|----------------|--------------|--------------|--------------|
> |     Madry                 |     83.56      |     56.69    |     51.92    |     51.00    |
> |     ADA-M + Linear        |     80.68      |     57.18    |     53.36    |     51.41    |
> |     ADA-M + Non-linear    |     80.42      |     57.98    |     54.44    |     52.51    |
> |     TRADES                |     81.39      |     57.25    |     53.64    |     51.39    |
> |     ADA-T + Linear        |     80.31      |     58.43    |     54.31    |     52.25    |
> |     ADA-T + Non-linear    |     81.22      |     58.97    |     54.55    |     52.95    |
>
> These results are evaluated on the best checkpoint models, which are adversarially trained on the CIFAR10 dataset.

---

> ### Author Response · Authors · 2021-11-19
> **Response to reviewer 65Tx [3/3]**
>
> Q3: “Can you provide examples of the kind of adversarial examples that ADA method is robust to, which prior work is not? That will be provide more intuition on what exactly is ADA method capturing. Right now, we see that the accuracy increases but not sure why.”
>
> A3:
> Thanks for your constructive suggestion! Inspired by your suggestion, we designed the following experiment to closely look at which kind of adversarial examples our method is robust to.
> To further analyze the superiority of our method, we compare ADA-M and Madry to explore which kind of adversarial examples that ADA-M is more robust to. These analyses are based on the spurious perspective, as one main conclusion of our paper is that the origin of adversarial vulnerability is the excessive focus of DNNs on spurious correlations between labels and style variables. Specifically, we calculate the KL-divergence between $KL(P(Y│h(x,s))||P(Y|h(x_adv,s)))$ for each input $x$, and divide samples into several (10 in our experiment) bins according to the KL-divergence. Then, we evaluate the robust accuracy of different models trained with ADA-M and Madry in each bin sample. We can see that ADA-M is more robust to samples leading to large KL-divergence than Madry. According to these empirical results, we can conclude that the proposed method is more robust to samples causing a significant difference between natural and adversarial distributions.
>
> |     KL       |     0.16    |     1.70    |     2.89    |     4.07    |     5.20    |     6.35    |     7.48    |     8.63    |     9.82    |     10.99    |     12.09    |
> |--------------|-------------|-------------|-------------|-------------|-------------|-------------|-------------|-------------|-------------|--------------|--------------|
> |     Madry    |     75.9    |     24.2    |     14.0    |     3.0     |     1.1     |     0.20    |     0.0     |     0.0     |     0.0     |     0.0      |     0.0      |
> |     ADA-M    |     73.5    |     38.2    |     30.8    |     24.7    |     18.9    |     17.3    |     12.0    |     10.6    |     7.4     |     1.8      |     0.0      |
>
> Again, thanks for your constructive suggestion. We have revised our paper following your suggestions. Some changes are:
>
> Sec. 3.2, Page 5: ‘To make sure that’.
>
> Appendix A.
>
> Appendix B, Table 5 and the corresponding explanation.
>
> Appendix F, Figure 3 and the corresponding explanation.

---

> > ### Comment · Reviewer_65Tx · 2021-11-30
> > **comments on the response**
> >
> > this is an interesting experiment. thanks for adding this result.

---

### Decision · Program_Chairs · 2022-01-20

**Decision:**

Accept (Poster)

**Comment:**

The paper shows a causal perspective to the adversarial robustness problem. Based on a causal graph of the adversarial data creation process, which describes the perceived data as a function of content and style variables, the authors identified that the spurious correlation between style and label is the main reason for adversarial examples. Based on this observation, they propose a method to learn models for which the conditional distribution of label given style and image does not vary much when attacked. Experiments on MNIST, CIFAR-10, and CIFAR-100 datasets show that the proposed method is better than two baselines, Madry and TRADES.

Overall, the paper contains interesting ideas and tackles an important problem. Due to some concerns regarding the clarity and motivation of the paper, we strongly recommend the authors take the reviewers' comments to heart and incorporate their thoughts in preparing the camera-ready version of their manuscript.